# Conversion of tropical forests to smallholder rubber and oil palm plantations impacts nutrient leaching losses and nutrient retention efficiency in highly weathered soils

Syahrul Kurniawan[1,3], Marife D. Corre[1*], Amanda L. Matson[1], Hubert Schulte-Bisping[2], Sri Rahayu Utami[3], Oliver van Straaten[1] and Edzo Veldkamp[1]

[1]Soil Science of Tropical and Subtropical Ecosystems, Faculty of Forest Sciences and Forest Ecology, University of Goettingen, Germany

[2]Soil Science of Temperate Ecosystems, Faculty of Forest Sciences and Forest Ecology, University of Goettingen, Germany

[3]Department of Soil Science, Faculty of Agriculture, Brawijaya University, Indonesia

**Correspondence to*: Marife D. Corre (mcorre@gwdg.de)

**Abstract.** Conversion of forest to rubber and oil palm plantations is widespread in Sumatra, Indonesia, and it is largely unknown how such land-use conversion affects nutrient leaching losses. Our study aimed to quantify nutrient leaching and nutrient retention efficiency in the soil after land-use conversion to smallholder rubber and oil palm plantations. In Jambi province, Indonesia, we selected two landscapes on highly weathered Acrisol soils that mainly differed in texture: loam and clay. Within each landscape, we compared two reference land uses: lowland forest and jungle rubber (defined as rubber trees interspersed in secondary forest) with two converted land uses, smallholder rubber and oil palm plantations. Within each landscape, the first three land uses were represented by four replicate sites and the oil palm by three sites, totaling to 30 sites. We measured leaching losses using suction cup lysimeters, sampled biweekly to monthly from February to December 2013. Forests and jungle rubber had low solute concentrations in drainage water, suggesting low internal inputs of rock-derived nutrients and efficient internal cycling of nutrients. These reference land uses on the clay Acrisol soils had lower leaching of dissolved N and base cations ($P$ = 0.01-0.06) and higher N and base cation retention efficiency ($P$ < 0.01-0.07) than those on the loam Acrisols. In the converted land uses, particularly on the loam Acrisol, the fertilized area of oil palm plantations showed higher leaching of dissolved N, organic C and base cations ($P$ < 0.01-0.08) and lower N and base cation retention efficiency compared to all the other land uses ($P$ < 0.01-0.06). The unfertilized rubber plantations, particularly on the loam Acrisol, showed lower leaching of dissolved P ($P$ = 0.08) and organic C ($P$ < 0.01) compared to forest or jungle rubber, reflecting decreases in soil P stocks and C inputs to the soil. Our results suggest that land-use conversion to rubber and oil palm causes disruption of initially efficient nutrient cycling, which decreases soil fertility. Over time, smallholders will likely be increasingly reliant on fertilization, with the risk of diminishing water quality due to increased nutrient leaching. Thus, there is a need to develop management practices to minimize leaching while sustaining productivity.

## 1 Introduction

Rainforests play an important role in maintaining ground water quality in tropical regions; however, in some regions their effectiveness may be decreasing as a consequence of forest conversion to agriculture. From 1990 to 2010, the deforestation rate in South and Southeast Asia was approximately 3 million ha $yr^{-1}$, of which 1.2 million ha $yr^{-1}$ occurred in Indonesia (FAO, 2010). During these two decades, the forest loss in the whole of Sumatra was 7.5 million ha, of which 1.1 million ha occurred in Jambi province (Margono et al., 2012). The two most common land uses replacing forests in Jambi province are oil palm and rubber plantations. From 2000 to 2010, the area of rubber plantations in Jambi increased by about 19% while oil palm plantations increased by 85% (Luskin et al., 2013). The expansion of rubber and oil palm plantations has increased the income of Jambi, in particular the smallholder farmers (Clough et al., 2016; Rist et al., 2010), as approximately 62% of oil palm landholdings in the Jambi Province are owned by smallholders (BPS, 2014). However, forest conversion to rubber and oil palm plantations has shown high ecological costs: losses in biodiversity (Clough et al., 2016), decreases in above- and below-ground organic carbon (C) stocks (Kotowska et al., 2015; van Straaten et al., 2015), reduction in soil nitrogen (N) availability (Allen et al., 2015), decrease in uptake of methane ($CH_4$) from the atmosphere into the soil (Hassler et al., 2015), and increase in soil $N_2O$ emission following N fertilization (Hassler et al., 2017).

Under similar climatic conditions and soil types, the two major factors that influence nutrient leaching losses from forest conversion are soil texture and management practices. Soil texture affects nutrient leaching through its control on soil fertility (e.g., cation exchange capacity, decomposition, and nutrient cycling) and soil water-holding capacity. Fine-textured soils have higher cation exchange capacity, decomposition and soil-N cycling rates, which result in higher soil fertility than coarse-textured soils (Allen et al., 2015; Silver et al., 2000; Sotta et al., 2008). Soil texture also influences water-holding capacity and drainage through its

effects on porosity, pore size distribution, and hydraulic conductivity (Hillel, 1982). Clay soils can hold a large amount of water and are dominated by small pores, which have low hydraulic conductivity in high moisture conditions. In contrast, coarse-textured soils have low water-holding capacity and are dominated by large pores, which conduct water rapidly in high moisture conditions, and therefore have high potential for leaching of dissolved solutes (Fujii et al., 2009; Lehman and Schroth, 2002). Thus, in heavily weathered soils, such as Acrisols, which dominate the converted lowland landscapes in Jambi, Indonesia **(**FAO et al., 2012), retention of their inherently low exchangeable base cations in the soil and maintenance of efficient soil-N cycling are largely influenced by soil texture (Allen et al., 2015).

Soil management practices (e.g., fertilizer and lime applications) in converted land uses also play an important role in influencing nutrient leaching, as the magnitude of dissolved nutrients moving downward with water is predominantly driven by the levels of those nutrients in the soil (Dechert et al., 2005, 2004). Without fertilization, nutrient leaching losses in agricultural land usually decrease with years following forest conversion (Dechert et al., 2004). This may be the case for the smallholder rubber plantations in our present study, as these have not been fertilized since conversion from forest (Allen et al., 2015; Hassler et al., 2017, 2015). However, soils in oil palm plantations are very often supplemented with chemical fertilizer and lime applications (Allen et al., 2015; Goh et al., 2003; Hassler et al., 2017, 2015). In cases where oil palm plantations are regularly fertilized, nutrient leaching losses in older plantations may be higher than in younger ones, as the applied nutrients accumulate in the subsoil over time (Goh et al., 2003; Omoti et al., 1983). Consequently, nutrient leaching in regularly fertilized oil palm plantations will likely be higher than in the original forest. Moreover, in our earlier study conducted in smallholder oil palm plantations, fertilization was shown to decrease microbial N immobilization due to decreases in microbial biomass (Allen et al., 2015), which could lead to decrease in retention of N in the soil.

Despite a growing body of information on the effects of deforestation on soil properties and processes, there is a lack of information on how forest conversion to rubber and oil palm influences nutrient leaching and the efficiency with which nutrients are retained in the soil. This lack is especially notable for nutrients other than N, as previous leaching studies commonly focus on this. Here, we present leaching losses of the full suite of major nutrients using a large-scale replicated design in a region affected by widespread land-use conversion to rubber and oil palm plantations. Our study aimed to assess: 1) how soil physical and biochemical characteristics affect nutrient leaching in highly weathered soils, and 2) the impact of land-use conversion to smallholder rubber and oil palm plantations on nutrient leaching and on N and base cation retention efficiency in the soil. We hypothesized that: 1) lowland forest and jungle rubber (rubber trees planted in secondary forest), which were the original land uses prior to conversion, will have lower leaching losses and higher nutrient retention in clay Acrisol soil than in loam Acrisol soil, and 2) smallholder oil palm plantations with fertilizer and lime applications will have the highest nutrient leaching losses (lowest nutrient retention) whereas smallholder rubber plantations with no fertilizer input will have the lowest nutrient leaching losses.

## 2 Materials and methods

### 2.1 Study sites and experimental design

Our study is part of the on-going multidisciplinary research project, EFForTS (http://www.uni-goettingen.de/en/310995.html), investigating the ecological and socioeconomic impact of conversion of lowland forest to rubber and oil palm plantations. The detailed experimental design and locations of the study sites were reported earlier (e.g., Allen et al., 2015; Hassler et al., 2017, 2015). In short, our study region is located in Jambi province, Indonesia (2° 0’ 57” S, 103° 15’ 33” E, 35 - 95 m elevation). The area has a mean annual air temperature of 26.7 ± 0.1

°C and a mean annual precipitation of 2235 ± 385 mm (1991–2011; data from a climate station at the Jambi Sultan Thaha airport from the Indonesian Meteorological, Climatological and Geophysical Agency). The dry season (<100 mm month$^{-1}$) is from May to September, and the wet season is from October to April. We selected two landscapes within our study region; while both were located on highly weathered Acrisol soils, one was a clay-textured soil and the other was a loam-textured soil (hereafter we refer to them as clay Acrisol and loam Acrisol landscapes). The soil textural difference leads to inherent differences in soil fertility, as shown by the higher effective cation exchange capacity, base saturation, Bray-extractable P and lower Al saturation in the clay than the loam Acrisols under forest and jungle rubber (Appendix Table A1; Allen et al. 2015). Within each landscape, we selected four land uses: lowland forest, jungle rubber, and smallholder plantations of rubber and oil palm (Appendix Table A2). Within each landscape, we had 15 sites (see Allen et al. 2015 for the map of these sites in the study region): four forest, four jungle rubber, four rubber plantations, and three oil palm plantations. We started with four oil palm sites at each landscape, but one plantation was sold and the new owner did not continue the collaboration with our research and in another site the instruments for leaching sampling were damaged. In our experimental design, land-use types (including the soil management practices typical for smallholders in the region) were the treatment and the sites were the replications. At each site, we established a plot of 50 m x 50 m. All plots were on the well-drained position of the landscape with slopes ranging from 3-10 % across all plots.

Based on our interviews with the smallholders, their plantations were established after clearing and burning of either forest or jungle rubber and hence these latter land uses served as the reference with which the converted plantations were compared. Additionally, the comparability of the initial soil conditions between the reference and converted land uses was tested using a land use-independent soil characteristic, i.e., clay content at 1–2 m depth (van Straaten et al., 2015); this did not statistically differ among land uses within each landscape

(Appendix Table A1; Allen et al., 2015; Hassler et al., 2015). Thus, changes in nutrient leaching can be attributed to land-use conversion with its inherent soil management practices. These first generation rubber and oil palm plantations were between 7 and 17 years of age. Tree density, height, basal area, and tree species abundance were higher in the reference land uses than the smallholder plantations (Appendix Table A2; Allen et al., 2015; Hassler et al., 2015; Kotowska et al., 2015).

Soil management practices in smallholder oil palm plantations are inherently varied (e.g., fertilization rate), as this depended on financial resources of the smallholders. Fertilization rates were 48 and 88 kg N $ha^{-1}$ $yr^{-1}$, 21 and 38 kg P $ha^{-1}$ $yr^{-1}$ and 40 and157 kg K $ha^{-1}$ $yr^{-1}$ (accompanied by Cl input of 143 kg Cl $ha^{-1}$ $yr^{-1}$) in the clay Acrisol and the loam Acrisol soils, respectively. Lime (e.g., $CaMg(CO_3)_2$), kieserite ($MgSO_4.H_2O$) and borate ($Na_2B_4O_2.5H_2O$) were also occasionally applied. These fertilization rates are typical of the smallholder farms in the region. Soil amendments were applied by hand around each palm tree at 0.8–1.5 m from the stem base. A combination of manual weeding and herbicides was practiced. Old oil palm fronds were regularly cut and stacked at 4–4.5 m from the palm rows (row spacing was about 9 m). The rubber plantations were not fertilized but were weeded both manually and with herbicides.

### 2.2 Lysimeter installation and soil water sampling

For measuring nutrient leaching, we sampled soil water using lysimeters, which were installed at two randomly chosen locations per replicate plot of the forest, jungle rubber and rubber plantations. In the oil palm plantations, the lysimeters were deployed according to the spatial structure of the soil management practices: one lysimeter was installed between 1.3–1.5-m distance from the tree stem where fertilizers were applied, and another lysimeter was installed between 4–4.5-m distance from the tree stem where the cut fronds were stacked. These suction cup lysimeters (P80 ceramic, maximum pore size 1 µm; CeramTec AG, Marktredwitz,

Germany) were inserted into the soil down to 1.5-m depth. This depth was based from our previous work in a lowland forest on highly weathered Ferralsol soil, where leaching losses were measured at various depth intervals down to 3 m and from which we found that leaching fluxes did not change below 1 m (Schwendenmann and Veldkamp, 2005). Moreover, this 1.5-m depth of lysimeter installation at our sites was well below the rooting depth, as determined from the fine-root biomass distribution with depths (Appendix Fig. B1; Kurniawan, 2016).

Prior to installation, lysimeters, tubes and collection containers were acid-washed and rinsed with deionized water. Lysimeters were installed in the field three months prior to the first sampling. The collection containers (dark glass bottles) were placed in plastic buckets with lids and buried in the ground approximately 2 m away from the lysimeters. Soil water was sampled biweekly to monthly, depending on the frequency of rainfall, from February to December 2013. Soil water was withdrawn by applying a 40 kPa vacuum on the sampling tube (Dechert et al., 2005; Schwendenmann and Veldkamp, 2005). The collected soil water was then transferred into clean 100-mL plastic bottles. Upon arrival at the field station, a subsample of 20 mL was set aside for pH measurement while the remaining sample was frozen. All frozen water samples were transported to the University of Goettingen, Germany and were kept frozen until analysis.

The total dissolved N (TDN), $NH_4^+$, $NO_3^-$ and $Cl^-$ concentrations were measured using continuous flow injection colorimetry (SEAL Analytical AA3, SEAL Analytical GmbH, Norderstedt, Germany). TDN was determined by ultraviolet-persulfate digestion followed by hydrazine sulfate reduction (Autoanalyzer Method G-157-96); $NH_4^+$ was analyzed by salicylate and dicloroisocyanuric acid reaction (Autoanalyzer Method G-102-93); $NO_3^-$ by cadmium reduction method with $NH_4Cl$ buffer (Autoanalyzer Method G-254-02); and $Cl^-$ was determined with an ion strength adjustor reagent that is pumped through an ion selective chloride electrode with an integrated reference electrode (Auto analyzer Method G-329-05). Dissolved organic N (DON) is the difference between TDN and mineral N ($NH_4^+$ + $NO_3^-$). Dissolved organic C

(DOC) was determined using a Total Organic Carbon Analyzer (TOC-Vwp, Shimadzu Europa GmbH, Duisburg, Germany). DOC was analyzed by pre-treating the samples with $H_3PO_4$ solution (to remove inorganic C) followed by ultraviolet-persulfate oxidation of organic C to $CO_2$, which is determined by an infrared detector. Base cations (Na, K, Ca, Mg), total Al, total Fe, total Mn, total S, total P, and total Si in soil water were analyzed using inductively coupled plasma-atomic emission spectrometer (iCAP 6300 Duo View ICP Spectrometer, Thermo Fischer Scientific GmbH, Dreieich, Germany). Instruments' detection limits were: 6 µg $NH_4^+$-N $L^{-1}$, 5 µg $NO_3^-$-N $L^{-1}$, 2 µg TDN $L^{-1}$, 4 µg DOC $L^{-1}$, 30 µg Na $L^{-1}$, 50 µg K $L^{-1}$, 3 µg Ca $L^{-1}$, 3 µg Mg $L^{-1}$, 2 µg Al $L^{-1}$, 3 µg Fe $L^{-1}$, 2 µg Mn $L^{-1}$, 10 µg P $L^{-1}$, 10 µg S $L^{-1}$, 1 µg Si $L^{-1}$ and 30 µg Cl $L^{-1}$.

Partial cation-anion charge balance of the major solutes (i.e., those with concentrations >0.03 mg $L^{-1}$) in soil water was done by expressing solute concentrations in $\mu mol_c$ $L^{-1}$ (molar concentration multiplied by the equivalent charge of each solute). Contributions of organic acids ($RCOO^-$) and bicarbonate ($HCO_3^-$) were calculated, together with S (having very low concentrations), from the difference between cations and anions. Charge contributions of total Al were assumed to be $3^+$, whereas solutes that had very low concentrations (i.e., total Fe, Mn and P), and thus had minimal charge contribution, as well as the total dissolved Si (commonly in a form of monosilicic acid ($H_4SiO_4^0$) that has no net charge) were excluded (similar to the method used by Hedin et al., 2003).

**2.3 Soil water modelling and calculation of nutrient leaching fluxes**

Drainage water fluxes were estimated using the soil water module of the Expert-N model (Priesack, 2005), which has been used in our earlier work on nutrient leaching losses in Sulawesi, Indonesia (Dechert et al., 2005). The model was parameterized with the characteristics measured at our sites, namely climate data, leaf area index, rooting depth, and soil characteristics. The climate variables included daily air temperature (minimum, maximum

and average), relative humidity, wind speed, solar radiation, and precipitation. For the loam Acrisol landscape, the climate data were taken from a climate station at the Harapan Forest Reserve, which was located 10–20 km from our sites. For the clay Acrisol landscape, the climate data were taken from the climate stations at the villages of Lubuk Kepayang and Sarolangun, which were respectively 10 km and 20 km from our sites. The leaf area indices measured in our forest, jungle rubber, rubber and oil palm sites in the loam Acrisol landscape were 5.8, 4.8, 3.5, and 3.9 $m^2$ $m^{-2}$, respectively, and in the clay Acrisol landscape were 6.2, 4.5, 2.8 and 3.1 $m^2$ $m^{-2}$, respectively (Rembold et al., unpublished data). Our measured fine root biomass distribution (Appendix Fig. B1; Kurniawan, 2016) was used to partition root water uptake at various soil depths. Soil characteristics included soil bulk density, texture (Appendix Table A1) and the water retention curve. The latter was determined using the pressure plate method for which intact soil cores (250 $cm^3$), taken at five soil depths (0.05, 0.2, 0.4, 0.75 and 1.25 m) from each land use within each landscape, were measured for water contents at pressure heads of 0, 100, 330 and 15000 hPa.

Calculation of drainage water fluxes followed the water balance equations:

$$\Delta W + D = P - R - ET \text{ and } ET = I + E + T$$

in which ΔW = change in soil water storage, D = drainage water below rooting zone, P = precipitation, R = runoff, ET = evapotranspiration, I = interception of water by plant foliage, E = evaporation from soil, and T = transpiration by plants. The Expert-N model calculates actual evapotranspiration using the Penman-Monteith method, runoff based on the sites' slopes, and vertical water movement using the Richards equation, of which the parameterization of the hydraulic functions were based on our measured soil texture and water retention curve (Mualem, 1976; Van Genuchten, 1980). To validate the output of the water model, we compared the modelled and measured soil matrix potential (Appendix Fig. B2). Soil matrix potential was measured biweekly to monthly from February to December 2013, using

tensiometers (P80 ceramic, maximum pore size 1 µm; CeramTec AG, Marktredwitz, Germany), which were installed at the depths of 0.3 m and 0.6 m in two replicate plots per land use within each landscape.

Modelled daily drainage water fluxes at a depth of 1.5 m were summed to get the biweekly or monthly drainage fluxes. Nutrient leaching fluxes were calculated by multiplying the element concentrations from each of the two lysimeters per replicate plot with the total biweekly or monthly drainage drainage water flux. The annual leaching flux was the sum of biweekly to monthly measured leaching fluxes from February to December 2013, added with the interpolated value for the unmeasured month of January 2013.

### 2.4 Nutrient retention efficiency

To evaluate the efficiency with which nutrients are retained in soil, we calculated the N and base cation retention efficiency as follows: 1 – (nutrient leaching loss/soil available nutrient) (Hoeft et al., 2014). For the oil palm plantations, we took the average leaching fluxes in the fertilized and frond-stacked areas of each plot for calculating the nutrient retention efficiency. This is because these sampling locations may contribute equally in terms of area as both the vertical and lateral flows in the soil profile could influence the sampled drainage water, and thus a wider area may contribute to the sampled drainage water than just the categorized sampling locations. For N retention efficiency calculation, TDN leaching flux was ratioed to gross N mineralization rate as the index of soil available N, with both terms expressed in mg N $m^{-2}$ $d^{-1}$. For calculation of base cation retention efficiency, base cation leaching flux was the sum of K, Na, Mg and Ca in units of $mol_{charge}$ $m^{-2}$ $yr^{-1}$ and soil available base cations was the sum of these exchangeable cations in units of $mol_{charge}$ $m^{-2}$. We used the measurements of gross N mineralization rate in the top 0.05-m depth and the stocks of exchangeable bases in the top 0.1-m depth (Appendix Table A1, reported by Allen et al., 2015).

### 2.5 Supporting parameter: nutrient inputs through bulk precipitation

In each landscape, we installed two rain samplers in an open area at 1.5 m above the ground. Rain samplers consisted of 1-liter high-density polyethylene bottles with lids attached to funnels that were covered with a 0.5-mm sieve, and were placed inside polyvinyl chloride tubes (to shield from sunlight and prevent algal growth). Rain samplers were washed with acid and rinsed with deionized water after each collection. Rain was sampled during the same sampling period as the soil water. Each rain sample was filtered through prewashed filter paper (4 μm pore size) into a 100 mL plastic bottle and stored frozen for transport to the University of Goettingen, Germany. The element analyses were the same as those described for soil water. The element concentrations in rainwater were weighted with the rainfall volume during the two-week or 1-month collection period to get volume-weighted concentrations. The annual element inputs from bulk precipitation were calculated by multiplying the volume-weighted average element concentrations in a year with the annual rainfall in each landscape.

### 2.6 Statistical analysis

Each replicate plot was represented by the average of two lysimeters, except for the oil palm plantations where lysimeters in fertilized and frond-stacked areas were analyzed separately. Tests for normality (Shapiro-Wilk's test) and homogeneity of variance (Levene's test) were conducted for each variable. Logarithmic or square-root transformation was used for variables that showed non-normal distribution and/or heterogeneous variance. We used linear mixed effects (LME) models (Crawley, 2009) to (1) assess differences between the two landscapes for the reference land uses (to answer objective 1), and (2) assess differences among land-use types within each landscape (to answer objective 2). The latter was analyzed for each landscape because the fertilization rates applied to the smallholder oil palm plantations inherently differed

between the two landscapes. For element concentrations, the LME model had landscape or land-use type as the fixed effect with spatial replication (plot) and time (biweekly or monthly measurements) as random effects. For the annual leaching fluxes, the LME model had landscape or land-use type as the fixed effect with spatial replication (plot) as a random effect. If they improved the relative goodness of the model fit (based on the Akaike information criterion), we extended the LME model to include (1) a variance function that allows different variances of the fixed effect, and/or (2) a first-order temporal autoregressive process that assumes that correlation between measurement periods decreases with increasing time intervals. Fixed effects were considered significant based on analysis of variance at $P \leq 0.05$, and differences between landscapes or land-use types were assessed using Fisher's least significant difference test at $P \leq 0.05$. Given the inherent spatial variability in our experimental design, we also considered $P$ values of $> 0.05 \leq 0.09$ as marginal significance, mentioned explicitly for some variables. To support the partial charge balance of dissolved cations and anions, we used Pearson correlation analysis to assess the relationships between solute cations and anions, using the monthly average ($n$ = 12) of the four replicate plots per land use within each landscape. We also used Pearson correlation analysis to test the modelled and measured soil matrix potential, using the monthly average ($n$ = 12) of the measured two replicate plots per land use within each landscape. To assess how the soil physical and biochemical characteristics (Table A1) influence the annual nutrient leaching fluxes, we conducted Spearman's rank correlation test for these variables, separately for the reference land uses and the converted land uses across landscapes ($n$ = 16). All statistical analyses were conducted using R 3.0.2 (R Development Core Team, 2013).

## 3 Results

### 3.1 Water balance and nutrient input from bulk precipitation

The modelled and measured soil matric potential were highly correlated ($R$ = 0.79 to 0.98, $n$ = 12, $P$ < 0.01) (Appendix Fig. B2). In forest and jungle rubber, modelled annual ET was 36-47 %, runoff was 16-27 %, and drainage was 32-44 % of annual precipitation (Table 1). In both landscapes, annual input from bulk precipitation was dominated by DOC (58 % of total element deposition), followed by Na, Cl, TDN, Ca, K and total S (Table 2). We compared the chlorinity ratios of elements in the bulk precipitation at our sites to those of seawater to infer anthropogenic influence. The average chlorinity ratios from both landscapes were 1.13 ± 0.05 for Na:Cl, 0.05 ± 0.01 for Mg:Cl, 0.20 ± 0.02 for Ca:Cl and 0.13 ± 0.04 for K:Cl, which were higher, except for Mg:Cl, than seawater chlorinity ratios (0.56 for Na:Cl, 0.07 for Mg:Cl, 0.02 for Ca:Cl and 0.02 for K:Cl; p. 349, Schlesinger and Bernhardt, 2013).

### 3.2 Element concentrations in soil water

For forest, the loam Acrisol had higher dissolved Na, Mg, total Al (all $P \leq 0.05$), $NH_4^+$-N, DON, total Fe and Cl concentrations (all $P \leq 0.09$) than the clay Acrisol (Table 3). For jungle rubber, the loam Acrisol had higher dissolved $NO_3^-$-N ($P \leq 0.05$) and lower total Si concentrations ($P \leq 0.09$) than the clay Acrisol (Table 3). The ionic charge concentration in soil solution of the forest sites was higher in the loam (274 ± 19 $\mu mol_{charge}\ l^{-1}$) than in the clay Acrisols (203 ± 20 $\mu mol_{charge}\ l^{-1}$) ($P$ = 0.01; Fig. 1), whereas in the jungle rubber these were comparable (loam Acrisols: 199 ± 31 $\mu mol_{charge}\ l^{-1}$, clay Acrisols: 207 ± 24 $\mu mol_{charge}\ l^{-1}$; Fig. 1). Correlation analysis of dissolved cations and anions in forest and jungle rubber showed that $NH_4^+$-N, Na, K, Ca, Mg and total Al were positively correlated with DON, DOC, Cl, $NO_3^-$-N and total S (Appendix Tables A3 and A4).

The rubber plantations in the loam Acrisol had lower $NO_3^-$-N, DON, DOC, Na, Ca, Cl (all $P \leq 0.05$), total P and total S concentrations (both $P \leq 0.08$) than either forest or jungle rubber (Table 3). This resulted in lower ionic charge concentration in soil solution of rubber plantation (200 ± 21 $\mu mol_{charge}\ l^{-1}$) than that of forest ($P < 0.01$; Fig. 1). In the clay Acrisol, only dissolved Na was lower in rubber plantations than in jungle rubber ($P \leq 0.01$; Table 3), and hence the ionic charge concentration in soil solution of rubber plantation (189 ± 23 $\mu mol_{charge}\ l^{-1}$) were comparable to those of the reference land uses (Fig. 1). In contrast to the reference land uses, unfertilized rubber plantations showed strong positive correlations of dissolved cations ($NH_4^+$-N, Na, K, Ca, Mg and total Al) with Cl and only weaker positive correlations with DOC or total S (Appendix Tables A3 and A4).

The fertilized areas of oil palm plantations had higher $NO_3^-$-N, Na, Ca, Mg, total Al, Cl (all $P \leq 0.05$) and lower soil solution pH ($P = 0.07$) than in the reference land uses within the loam Acrisol landscape (Table 3). In the clay Acrisol landscape, the fertilized areas of oil palm plantations had higher soil solution pH and dissolved Na (both $P \leq 0.05$) whereas DON was lower ($P = 0.08$) than the reference land uses (Table 3). Ionic charge concentrations in soil solutions of the fertilized areas of oil palm plantations (648 ± 306 $\mu mol_{charge}\ l^{-1}$ for loam Acrisol and 317 ± 83 $\mu mol_{charge}\ l^{-1}$ for clay Acrisol) were higher than in frond-stacked areas (190 ± 23 $\mu mol_{charge}\ l^{-1}$ for loam Acrisol and 173 ± 37 $\mu mol_{charge}\ l^{-1}$ for clay Acrisol) and in other land uses ($P < 0.01$; Fig. 1). In the fertilized areas of the loam Acrisol, dissolved $NO_3^-$-N was positively correlated with total Al (Table A3) and both were negatively correlated with soil solution pH ($R$ = -0.57 to -0.76, $n$ = 12, $P \leq 0.05$). The fertilized areas showed strong positive correlations of dissolved cations (Na, K, Ca, Mg and total Al) with total S or Cl and only weaker positive correlations with DOC (Appendix Tables A3 and A4). The frond-stacked areas showed positive correlations of these dissolved cations largely with Cl (Appendix Tables A3 and A4).

### 3.3 Annual leaching flux and nutrient retention efficiency

For forest, annual leaching fluxes of Na, Ca, Mg, total Al, Cl (all $P \leq 0.05$), $NH_4^+$-N, DON, total Si ($P \leq 0.09$) were larger in the loam than in the clay Acrisols, whereas in jungle rubber only annual $NO_3^-$-N leaching flux was larger ($P \leq 0.05$) (Table 4). Across all forest and jungle rubber sites, annual leaching fluxes of anions (DON and $NO_3^-$-N) were negatively correlated with indicators of soil exchangeable cations (base saturation, effective cation exchange capacity (ECEC), exchangeable Al; *Spearman's* $\rho$ = -0.51 to -0.61, $n$ = 16, $P \leq 0.05$), while annual $NH_4^+$-N leaching flux was negatively correlated (*Spearman's* $\rho$ = -0.53, $n$ = 16, $P$ = 0.04) with soil organic C (Table A1). For both reference land uses, the higher leaching in loam than in clay Acrisols was mirrored by decreases in N and base cation retention efficiency in the soil (Table 5). Across all reference sites, N and base cation retention efficiency in the soil were positively correlated with base saturation, ECEC and soil organic C (*Spearman's* $\rho$ = 0.52 to 0.70, $n$ = 16, $P \leq 0.04$) which, in turn, were positively correlated with clay content (*Spearman's* $\rho$ = 0.55 to 0.59, $n$ = 12 sites analyzed for clay content, $P \leq 0.05$).

The rubber plantations had lower annual P leaching flux than forests ($P$ = 0.08) and lower annual DOC leaching flux than jungle rubber in the loam Acrisol ($P < 0.01$) (Table 4). N and base cation retention efficiency in the soil of rubber plantations were comparable with the reference land uses in both landscapes (Table 5). In oil palm plantations of the loam Acrisol landscape, the fertilized areas had higher annual leaching fluxes of $NO_3^-$, TDN, DOC, Na, Ca, Mg, total Al, total S and Cl (all $P \leq 0.05$) than in the unfertilized rubber plantations or the reference land uses, whereas the frond-stacked areas showed comparable leaching fluxes with the other land uses (Table 4). In the loam Acrisol, oil palm plantations had lower N and base cation retention efficiency in the soil than the other land uses ($P \leq 0.01 – 0.06$; Table 5). In the clay Acrisol landscape, where leaching fluxes were small (Table 4), there were no differences observed in soil N and base cation retention efficiency among land uses (Table 5). Across all

rubber and oil palm sites, annual $NH_4^+$-N and DON leaching fluxes were negatively correlated with ECEC and clay content (*Spearman's* $\rho$ = -0.50 to -0.64, $n \leq 16$, $P$ = 0.03 – 0.07). Moreover, base cation retention efficiency in the soil was positively correlated with ECEC, soil organic C and clay content (*Spearman's* $\rho$ = 0.68 to 0.91, $n \leq 16$, $P \leq 0.01 – 0.02$) which, in turn, were correlated with each other (*Spearman's* $\rho$ = 0.87 to 0.90, $n$ = 12 sites analyzed for clay content, $P \leq 0.01$).

## 4 Discussion

### 4.1 Water balance and nutrient input from bulk precipitation

Our modelled water balance was generally comparable with the estimates from other studies in Indonesia. When compared to a forest at 200-500 m elevation on a clay loam soil in Kalimantan (with 28-47 % ET and 40-55 % runoff of 3451 mm $yr^{-1}$ precipitation; Suryatmojo et al., 2013), our estimated ET in the forest sites was comparable, although our modelled runoff was lower (Table 1). However, our runoff estimates were similar to the modelled runoff in oil palm and rubber plantations in Jambi province (10-20 % of rainfall; Tarigan et al., 2016). Our values for runoff and drainage flux in oil palm plantations (Table 1) were similar to oil palm plantations at 130 m elevation on Andisol soils in Papua New Guinea (with 37-57 % ET, 0-44 % runoff, and 38-59 % drainage of 2398-3657 mm $yr^{-1}$ precipitation; Banabas et al., 2008). Additionally, our estimated daily ET in oil palm (2.4 ± 0.1 and 2.2 ± 0.1 mm $d^{-1}$ in the loam and clay Acrisols, respectively) was similar to the measurements of Niu et al. (2015) (2.6 ± 0.7 mm $d^{-1}$) in the same oil palm plantations included in our study. Finally, the high correlations between modelled and measured matric potential (0.3-m depth; Appendix Fig. B2) suggest that our modelled drainage fluxes closely approximated those in the studied land uses.

The chemical composition of bulk precipitation in our study area was clearly influenced by anthropogenic activities, likely from biomass burning and/or terrigenous dust from

agriculture. This is evident from the high DOC and TDN, which were comparable to values from bulk precipitation impacted by biomass burning in Brazil, Panama and Costa Rica (Coelho et al., 2008; Corre et al., 2010; Eklund et al., 1997). The high Na:Cl, K:Cl and Ca:Cl ratios in bulk precipitation at our sites were similar to values from bulk precipitation influenced by dusts in Singapore and Costa Rica (Balasubramanian et al., 1999; Eklund et al., 1997). The N deposition from bulk precipitation (Table 2) was only 0.7-1.4 % of the gross rate of N mineralization in the top 0.05 m of soil at our forest sites (250-600 mg N $m^{-2}$ $d^{-1}$; Allen et al., 2015). The amount of P and base cations from bulk precipitation (Table 2) were also only 1-3 % of the stocks of Bray-extractable P and exchangeable base cations in the top 0.1 m of soil at our forest sites (1-4 g P $m^{-2}$, 22-65 g K $m^{-2}$, 57-109 g Ca $m^{-2}$, and 8-29 g Mg $m^{-2}$; Allen et al., 2016). Thus, in our study area, the much larger stocks and cycling rates of nutrients in the soil (and how these are affected by land-use change) will be a more significant influence on nutrient leaching losses (see below) than the low amounts of nutrients from bulk precipitation.

**4.2 Forest and jungle rubber: leaching fluxes and nutrient retention efficiency**

The Acrisol soils in our study region exhibited similarly low ionic charge concentration and high dissolved Al in soil solutions of forest and jungle rubber (Fig. 1) as those reported for drainage and stream waters in highly weathered Ferralsol soils (Hedin et al., 2003; Markewitz et al., 2001). Low solute concentration in soil solution of highly weathered soils is due to minimal internal input of rock-derived nutrients via weathering (Hedin et al., 2003). Soil fertility of such highly weathered soils is conserved through efficient cycling of nutrients between the soil and vegetation, for which soil texture is one important controlling factor. Previous studies have shown that fine-textured, highly weathered Acrisol and Ferralsol soils have higher nutrient- and water-holding capacity, higher soil N availability, decomposition rate and plant productivity than coarse-textured Acrisols and Ferralsol soils (e.g., Ohta et al., 1993;

Silver et al., 2000; Sotta et al., 2008). Our measured nutrient leaching losses from the reference land uses supported these findings. The lower solute concentrations (Table 3) and lower annual nutrient leaching fluxes in clay as compared to loam Acrisols (i.e., TDN, Na, Ca, Mg; Table 4) were paralleled by higher rates of soil $NH_4^+$ cycling (Allen et al., 2015), higher soil N stocks, ECEC and base saturation (Appendix Table A1), higher water-holding capacity (Hassler et al., 2015) and lower drainage fluxes (Table 1). Since leaching of DON and mineral N was associated with leaching of dissolved cations (Appendix Tables A3 and A4), high rates of soil-N cycling in the clay Acrisol (Allen et al., 2015) had contributed to lower leaching of N with base cations, and thus conserving soil fertility (Appendix Table A1).

We also observed a link between N leaching and the acid-buffering capacity of the soils, as shown by the negative correlations of annual DON and $NO_3^-$-N leaching losses with soil base saturation, ECEC and exchangeable Al. The higher the N and cation leaching (as in the loam Acrisol), the lower were the cation stocks and ECEC in the soil (Appendix Table A1). Similarly, the negative correlation of annual $NH_4^+$-N leaching losses with soil organic C suggest high retention of $NH_4^+$ in the clay Acrisol that has higher soil organic C (Appendix Table A1), higher soil microbial biomass and higher gross rate of $NH_4^+$ cycling than in the loam Acrisol (Allen et al., 2015). These all led to the higher N and base cation retention efficiency in clay than in loam Acrisols (Table 5), reflecting the higher nutrient- and water-retention capacity of the clay Acrisols. The positive correlations of N and base cation retention efficiency with soil base saturation, ECEC, organic C and clay content suggest efficient cycling of nutrients between soil and vegetation in the clay Acrisol. In summary, our findings showed that soil texture was an important factor regulating nutrient leaching losses and soil fertility in these highly weathered Acrisol landscapes.

**4.3. Leaching fluxes in rubber**

The low ionic charge concentration in soil solutions of unfertilized rubber plantations, particularly in the loam Acrisol (Fig. 1; Table 3), reflected decreased leaching losses after 14-17 yrs of land-use conversion (Appendix Table A2). Land-use conversion by smallholders entails slashing and burning of the original vegetation as well as localized manual cultivation. A large portion of nutrients in biomass are lost during burning (Kaufmann et al., 1995; Mackensen et al., 1996) and the pulse release of nutrients from ashes and decomposition results in high nutrient leaching losses immediately after burning followed by continuous decreases in leaching losses with time (Klinge et al., 2004).

Previous studies have shown that soil nutrient levels decrease significantly after years of agricultural production without soil amendments, e.g., decreases in exchangeable bases (Dechert et al. 2004), P availability (Ngoze et al., 2008), and soil N availability (Allen et al., 2015; Corre et al., 2006; Davidson et al., 2007). This was also evident in our unfertilized rubber plantations where, in the loam Acrisol, we measured lower annual P and DOC leaching fluxes than either in forest or jungle rubber (Table 4). The decrease in annual P leaching flux was reflected by a decrease in Bray-extractable P in the entire 2-m soil depth of the same rubber plantations compared to forest (Allen et al., 2016). Similarly, the decrease in annual DOC leaching flux was mirrored by the decreases in microbial C (Allen et al., 2015), litterfall and root production (Kotowska et al., 2015) in the same rubber plantations, and the overall decrease in soil organic C stocks in smallholder rubber plantations in the same study region (van Straaten et al., 2015) compared to forest. Decreases in DOC concentrations of soil solutions were possibly the reason why cations in the soil solutions of the rubber plantations were strongly correlated with Cl and only weakly correlated with organic-associated anions (DOC or total S; Appendix Tables A3 and A4). Our results showed that disruption of nutrient cycling between the soil and original vegetation brought about by land-use conversion to rubber plantations,

combined with the absence of soil amendments, had decreased P and DOC leaching which suggest a decrease in soil fertility.

**4.4 Leaching fluxes in oil palm and nutrient retention efficiency in converted land uses**

The most important factor influencing nutrient leaching in the smallholder oil palm plantations was fertilizer application. This was evident by the higher solute concentrations of the fertilized area compared to the frond-stacked area and to the other land uses (Fig. 1; Table 3). In the fertilized area, the stronger correlations of dissolved cations with total S and Cl, rather than with DOC, were because S and Cl are components of the applied fertilizers (see 2.1). The larger increases in solute concentrations in fertilized area of the loam Acrisol compared to fertilized area of the clay Acrisol were due to the following: 1) higher fertilization rates of oil palm plantations in the loam Acrisol landscape (see 2.1), and 2) its lower clay content that contributed to its lower nutrient- and water-holding capacity (Appendix Table A1; Table 1). In fertilized areas of the loam Acrisol, the correlations among dissolved $NO_3^-$, total Al and acidity were likely due to nitrification of added N fertilizer and the low acid-buffering capacity of this loam Acrisol soil (i.e., low base saturation in the top 0.1 m (Appendix Table A1) and in the entire 2-m depth; Allen et al. 2016). Soil extractable $NO_3^-$ and $NH_4^+$ in these smallholder plantations are elevated up to six weeks following fertilization (Hassler et al., 2017), during which time $NO_3^-$ is susceptible to leaching. Nitrification-induced acidity may have enhanced the Al acid-buffering reaction and led to the increases in dissolved Al and acidity of soil solution. Other studies in Indonesia and Malaysia have also reported increases in soil acidity due to N fertilization in oil palm plantations (Anuar et al., 2008; Comte et al., 2013). Even though occasional liming is practiced by smallholders in these oil palm plantations, soil pH (Appendix Table A1) was still within the Al acid-buffering range (pH 3-5; Van Breemen et al., 1983). The acidic soil water and elevated dissolved Al concentration resulting from N fertilization in these

oil palm plantations may also have triggered the decrease in mycorrhizal colonization of fine roots and the increase in distorted root tips found at the same sites (Sahner et al., 2015).

In the fertilized areas of oil palm plantations in the loam Acrisol, increased annual leaching fluxes of Na, total S, Cl, $NO_3^-$, TDN, Ca and Mg (Table 4) were due to applications of Na-, S- and N-containing fertilizers and lime (see 2.1). The leaching losses in our oil palm plantations were lower than those reported for oil palm plantations on Acrisol soils in Nigeria (2.6 g Ca $m^{-2}$ and 0.6 g Mg $m^{-2}$ during a six-month period; Omoti et al., 1983) and Malaysia (0.3-0.6 g N $m^{-2}$ during a five-month period; Tung et al., 2009), and on Andisol soils in Papua New Guinea (3.7-10.3 g N $m^{-2}$ $yr^{-1}$ during a fourteen-month period; Banabas et al., 2008), all of which had larger fertilization rates than our smallholders. Moreover, the increased annual DOC fluxes in fertilized areas of oil palm plantations (Table 4) suggests a reduction in the retention of DOC in the soil. When combined with the decreases in litterfall and root production, harvest export (Kotowska et al., 2015), and decreases in soil $CO_2$ emissions (Hassler et al., 2015) from the same oil palm plantations, this provides strong support for the decreases in soil organic C stocks in smallholder oil palm plantations in the same study region (van Straaten et al., 2015).

Altogether, our results showed the overarching influence of soil texture on nutrient- and water-holding capacity in these converted land uses. First, this was evident from the increased leaching of TDN and base cations, particularly in the loam Acrisol, in fertilized oil palm plantations (Table 4) that led to decreased N and base cation retention efficiency in the soil (Table 5). Second, this was shown by the positive correlations of annual $NH_4^+$-N leaching, annual DON leaching and base cation retention efficiency with ECEC, soil organic C and clay content across all sites of the converted land uses.

## 5 Conclusions

The low solute concentrations in drainage water of the reference land uses signified low internal inputs of rock-derived nutrients in these highly-weathered soils, and suggest efficient internal cycling of nutrients. Our findings of lower nutrient leaching losses and higher nutrient retention efficiency in the reference land uses on the clay as compared to the loam Acrisol soils supported our first hypothesis, and reflected the influence of soil texture on nutrient- and water-holding capacity. The low nutrient leaching losses in the unfertilized rubber plantations and the high leaching in the fertilized oil palm plantations supported our second hypothesis. Reduced P and DOC leaching in rubber plantations signaled reduction in soil fertility, which may influence how long these rubber plantations can remain before conversion to another land use. Sustainability of oil palm plantations must take into account the long-term effect of chronic N fertilization on soil water acidity and Al solubility; the inherently low acid-buffering capacity of Acrisol soils implies that the smallholders will be increasingly dependent on lime application, which entails additional capital input. Our results highlight the need to develop soil management practices that conserve soil fertility in unfertilized rubber plantations and increase nutrient retention efficiency in fertilized oil palm plantations, in order to minimize the reductions of ecosystem provisioning services (e.g., soil fertility and water quality) and hinder further forest conversion.

Further quantification of leaching losses should focus on large-scale oil palm plantations, which have 2-3 times higher fertilization rates than the smallholder plantations, as they may have a larger impact on ground water quality than the smallholder plantations. For valid large-scale extrapolation, quantification of leaching losses in oil palm plantations should not only represent the spatial structure of management practices but also surface landforms, which influence water redistribution (e.g., inclusion of riparian areas), and an improved water budget (e.g., estimates of evapotranspiration from inter-rows).

*Data availability*

Our data are deposited in the EFForTS-IS data repository (https://efforts-is.uni-goettingen.de), an internal data-exchange platform, which is accessible to EFForTS members only. Based on data sharing agreement within EFForTS, these data are currently not publicly accessible but will be made available through a written request to the corresponding and senior authors.

*Author contribution*

SK, MDC, EV and SRU conceived and designed research. SK carried out field measurements. MDC, EV and SRU supported the field research. SK and HSB modelled water budget with the Expert N water module. SK, MDC and EV analyzed the data. SK, MDC, ALM, OvS and EV wrote the manuscript.

*Competing interests*

All authors declare no conflict of interest.

*Acknowledgements*. This study was funded by the Deutsche Forschungsgemeinschaft (DFG) as part of subproject A05 (SFB 990/2) in the Collaborative Research Center 990 (EFForTS). Kurniawan received a postgraduate scholarship from the Indonesian Directorate General of Higher Education. We thank the village leaders, smallholders, PT REKI and Bukit Duabelas National Park for fruitful collaboration. We are especially grateful to our Indonesian assistants and the rangers of the forest areas. We acknowledge the Indonesian Meteorological, Climatological and Geophysical Agency and the subprojects A03 and B06 for data sharing. We also thank Andrea Bauer, Dirk Böttger, Martina Knaust and Kerstin Langs for their assistance. This study was conducted using the research permits 215/SIP/FRP/SM/VI/2012 and 44/EXT/SIP/FRP/SM/V/2013, and the collection permits 2703/IPH.1/KS.02/XI/2012 and S.13/KKH-2/2013, recommended by RISTEK and LIPI and issued by PHKA, Indonesia.

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

**Table 1.** Simulated water balance during 2013 in different land uses within two landscapes
(loam and clay Acrisol soils) in Jambi, Sumatra, Indonesia.

| Water balance components (mm yr$^{-1}$) | Forest | Jungle rubber | Rubber plantations | Oil palm plantations |
|---|---|---|---|---|
| | loam Acrisol soil (precipitation: 3418 mm yr$^{-1}$) | | | |
| Evapotranspiration | 1384 | 1224 | 1077 | 1027 |
| Transpiration | 1033 | 815 | 594 | 437 |
| Evaporation | 155 | 213 | 287 | 408 |
| Interception | 196 | 196 | 196 | 182 |
| Water drainage | 1483 | 1487 | 1544 | 1614 |
| Runoff | 545 | 704 | 800 | 761 |
| | clay Acrisol soil (precipitation: 3475 mm yr$^{-1}$) | | | |
| Evapotranspiration | 1622 | 1271 | 1114 | 1071 |
| Transpiration | 1284 | 861 | 402 | 446 |
| Evaporation | 157 | 242 | 548 | 459 |
| Interception | 181 | 168 | 164 | 166 |
| Water drainage | 1117 | 1268 | 1280 | 1311 |
| Runoff | 722 | 932 | 1070 | 1087 |

**Table 2.** Mean (± SE, *n* = 2) volume-weighted element concentrations and annual inputs in bulk precipitation, measured bi-weekly to monthly from February to December 2013 within two landscapes (loam and clay Acrisol soils) in Jambi, Sumatra, Indonesia.

| Elements | Volume-weighted concentration (mg $l^{-1}$) | | Annual input (g $m^{-2}$ $yr^{-1}$) | |
|---|---|---|---|---|
| | loam Acrisol | clay Acrisol | loam Acrisol | clay Acrisol |
| Ammonium ($NH_4^+$-N) | 0.17 (0.02) | 0.20 (0.02) | 0.58 (0.06) | 0.69 (0.07) |
| Nitrate ($NO_3^-$-N) | 0.04 (0.02) | 0.07 (0.01) | 0.13 (0.06) | 0.26 (0.04) |
| Dissolved organic nitrogen (N) | 0.17 (0.01) | 0.20 (0.04) | 0.58 (0.02) | 0.70 (0.14) |
| Total dissolved nitrogen (N) | 0.38 (0.00) | 0.47 (0.07) | 1.29 (0.01) | 1.64 (0.26) |
| Dissolved organic carbon (C) | 8.15 (0.19) | 7.44 (0.07) | 27.84 (0.66) | 25.86 (0.25) |
| Sodium (Na) | 1.84 (0.04) | 1.90 (0.18) | 6.30 (0.13) | 6.61 (0.63) |
| Potassium (K) | 0.16 (0.04) | 0.28 (0.14) | 0.55 (0.15) | 0.96 (0.49) |
| Calcium (Ca) | 0.32 (0.02) | 0.36 (0.07) | 1.09 (0.08) | 1.24 (0.24) |
| Magnesium (Mg) | 0.07 (0.01) | 0.09 (0.01) | 0.24 (0.05) | 0.30 (0.04) |
| Total aluminum (Al) | 0.02 (0.01) | 0.01 (0.00) | 0.05 (0.03) | 0.04 (0.01) |
| Total iron (Fe) | 0.01 (0.00) | 0.01 (0.00) | 0.04 (0.01) | 0.03 (0.01) |
| Total manganese (Mn) | 0.001 (0.00) | 0.001 (0.00) | 0.003 (0.00) | 0.004 (0.00) |
| Total phosphorus (P) | 0.01 (0.00) | 0.02 (0.00) | 0.04 (0.01) | 0.08 (0.01) |
| Total sulfur (S) | 0.26 (0.00) | 0.30 (0.03) | 0.90 (0.01) | 1.04 (0.10) |
| Total silica (Si) | 0.02 (0.01) | 0.03 (0.01) | 0.06 (0.02) | 0.09 (0.03) |
| Chloride (Cl) | 1.79 (0.25) | 1.54 (0.30) | 6.11 (0.84) | 5.34 (1.06) |

**Table 3.** Mean (± SE, $n$ = 4, except for oil palm $n$ = 3) nutrient concentrations in soil solution from a depth of 1.5 m in different land uses within two landscapes (loam and clay Acrisol soils) in Jambi, Sumatra, Indonesia. Means followed by different lowercase letters indicate significant differences among land uses within each landscape and different uppercase letters indicate significant differences between landscapes for each reference land use (Linear mixed effects models with Fisher's LSD test at $P \leq 0.05$, and † at $P \leq 0.09$ for marginal significance).

| Elements | Forest | Jungle rubber | Rubber | Oil palm fertilized area | Oil palm frond-stacked area |
|---|---|---|---|---|---|
| | | | loam Acrisol soil | | |
| pH | 4.3 (0.0) a† | 4.3 (0.1) a† | 4.4 (0.0) a† | 4.1 (0.1) b† | 4.3 (0.0) a† |
| Ammonium (mg $NH_4^+$-N $l^{-1}$) | 0.2 (0.0) A† | 0.3 (0.1) | 0.2 (0.0) | 0.2 (0.0) | 0.2 (0.0) |
| Nitrate (mg $NO_3^-$-N $l^{-1}$) | 0.1 (0.1) b | 0.1 (0.0) b A | 0.0 (0.0) c | 0.3 (0.2) a | 0.1 (0.0) b |
| Dissolved organic N (mg N $l^{-1}$) | 0.2 (0.0) a A† | 0.1 (0.0) b | 0.1 (0.0) b | 0.1 (0.0) ab | 0.1 (0.0) b |
| Total dissolved N (mg N $l^{-1}$) | 0.5 (0.1) A† | 0.4 (0.1) A† | 0.2 (0.0) | 0.6 (0.2) | 0.3 (0.0) |
| Dissolved organic C (mg C $l^{-1}$) | 3.7 (0.3) ab | 4.0 (0.5) ab | 3.1 (0.2) c | 4.2 (0.1) a | 3.6 (0.1) b |
| Sodium (mg Na $l^{-1}$) | 3.2 (0.1) b A | 2.4 (0.2) c | 2.2 (0.2) c | 7.2 (3.9) a | 2.3 (0.3) c |
| Potassium (mg K $l^{-1}$) | 0.4 (0.0) | 0.2 (0.1) | 0.3 (0.1) | 0.4 (0.1) | 0.4 (0.1) |
| Calcium (mg Ca $l^{-1}$) | 0.8 (0.0) b | 0.7 (0.1) c | 0.7 (0.1) c | 2.7 (0.9) a | 0.7 (0.1) c |
| Magnesium (mg Mg $l^{-1}$) | 0.3 (0.0) b A | 0.2 (0.0) c | 0.3 (0.1) b | 0.5 (0.1) a | 0.2 (0.0) c |

| | | | | | |
|---|---|---|---|---|---|
| Total aluminum (mg Al $l^{-1}$) | 0.4 (0.1) $_{b\ A}$ | 0.2 (0.0) $_{c}$ | 0.3 (0.0) $_{b}$ | 1.2 (0.7) $_{a}$ | 0.1 (0.0) $_{c}$ |
| Total iron (mg Fe $l^{-1}$) | 0.2 (0.1) $_{A†}$ | 0.0 (0.0) | 0.0 (0.0) | 0.0 (0.0) | 0.1 (0.1) |
| Total manganese (mg Mn $l^{-1}$) | 0.02 (0.00) | 0.01 (0.00) | 0.01 (0.00) | 0.01 (0.00) | 0.01 (0.00) $_{B}$ |
| Total phosphorus (mg P $l^{-1}$) | 0.008 (0.0) $_{a†}$ | 0.004 (0.0) $_{b†}$ | 0.003 (0.0) $_{c†}$ | 0.005 (0.0) $_{ab†}$ | 0.005 (0.0) $_{ab†}$ |
| Total sulfur (mg S $l^{-1}$) | 0.16 (0.00) $_{a†}$ | 0.14 (0.00) $_{bc†}$ | 0.10 (0.00) $_{c†}$ | 0.14 (0.00) $_{ab†}$ | 0.12 (0.00) $_{b†}$ |
| Total silica (mg Si $l^{-1}$) | 0.5 (0.1) | 0.3 (0.1) $_{B†}$ | 0.2 (0.1) | 0.3 (0.1) | 0.2 (0.0) |
| Chloride (mg Cl $l^{-1}$) | 8.9 (0.8) $_{b\ A†}$ | 6.6 (0.8) $_{c}$ | 6.7 (0.6) $_{c}$ | 21.0 (2.7) $_{a}$ | 6.2 (0.8) $_{c}$ |
| | | | clay Acrisol soil | | |
| pH | 4.3 (0.1) $_{c}$ | 4.4 (0.1) $_{bc}$ | 4.4 (0.0) $_{c}$ | 4.6 (0.1) $_{ab}$ | 4.6 (0.1) $_{a}$ |
| Ammonium (mg $NH_4^+$-N $l^{-1}$) | 0.2 (0.0) $_{B†}$ | 0.1 (0.0) | 0.1 (0.0) | 0.2 (0.0) | 0.1 (0.0) |
| Nitrate (mg $NO_3^-$-N $l^{-1}$) | 0.1 (0.0) | 0.0 (0.0) $_{B}$ | 0.2 (0.1) | 0.9 (0.9) | 0.0 (0.0) |
| Dissolved organic N (mg N $l^{-1}$) | 0.1 (0.0) $_{a†B†}$ | 0.1 (0.0) $_{a†}$ | 0.1 (0.0) $_{ab†}$ | 0.0 (0.0) $_{b†}$ | 0.0 (0.0) $_{b†}$ |
| Total dissolved N (mg N $l^{-1}$) | 0.3 (0.0) $_{B†}$ | 0.2 (0.0) $_{B†}$ | 0.4 (0.1) | 1.1 (0.9) | 0.2 (0.0) |
| Dissolved organic C (mg C $l^{-1}$) | 3.3 (0.4) | 4.0 (0.3) | 2.9 (0.1) | 4.8 (0.9) | 4.4 (1.1) |
| Sodium (mg Na $l^{-1}$) | 2.4 (0.2) $_{bc\ B}$ | 2.5 (0.1) $_{b}$ | 2.0 (0.1) $_{c}$ | 4.6 (1.2) $_{a}$ | 2.5 (0.5) $_{bc}$ |
| Potassium (mg K $l^{-1}$) | 0.3 (0.0) | 0.3 (0.1) | 0.3 (0.0) | 0.4 (0.1) | 0.2 (0.1) |
| Calcium (mg Ca $l^{-1}$) | 0.7 (0.1) | 0.7 (0.0) | 0.7 (0.1) | 0.8 (0.2) | 0.5 (0.1) |

| | | | | | |
|---|---|---|---|---|---|
| Magnesium (mg Mg $l^{-1}$) | 0.3 (0.0) B | 0.3 (0.0) | 0.3 (0.0) | 0.4 (0.1) | 0.2 (0.1) |
| Total aluminum (mg Al $l^{-1}$) | 0.2 (0.0) B | 0.2 (0.1) | 0.3 (0.1) | 0.2 (0.1) | 0.1 (0.0) |
| Total iron (mg Fe $l^{-1}$) | 0.0 (0.0) b† B† | 0.0 (0.0) b† | 0.0 (0.0) b† | 0.0 (0.0) b† | 0.1 (0.0) a† |
| Total manganese (mg Mn $l^{-1}$) | 0.01 (0.00) | 0.01 (0.00) | 0.01 (0.00) | 0.08 (0.10) | 0.02 (0.00) |
| Total phosphorus (mg P $l^{-1}$) | 0.010 (0.0) | 0.004 (0.0) | 0.004 (0.0) | 0.004 (0.0) | 0.010 (0.0) |
| Total sulfur (mg S $l^{-1}$) | 0.15 (0.00) | 0.11 (0.00) | 0.11 (0.00) | 0.13 (0.00) | 0.12 (0.00) |
| Total silica (mg Si $l^{-1}$) | 0.4 (0.0) | 0.6 (0.1) A† | 0.3 (0.0) | 1.0 (0.4) | 0.7 (0.2) |
| Chloride (mg Cl $l^{-1}$) | 6.4 (0.6) B† | 6.8 (0.9) | 5.7 (0.8) | 7.2 (2.1) | 4.6 (0.8) |

**Table 4.** Mean (± SE, *n* = 4, except for oil palm *n* = 3) annual (2013) nutrient leaching fluxes measured at a depth of 1.5 m in different land uses within two landscapes (loam and clay Acrisol soils) in Jambi, Sumatra, Indonesia. Means followed by different lowercase letters indicate significant differences among land uses within each landscape and different uppercase letters indicate significant differences between landscapes for each reference land use (Linear mixed effects models with Fisher's LSD test at $P \leq 0.05$, and † at $P \leq 0.09$ for marginal significance).

| Elements | Forest | Jungle rubber | Rubber | Oil palm fertilized area | Oil palm frond-stacked area |
|---|---|---|---|---|---|
| | | loam Acrisol soil | | | |
| Ammonium ($g\ NH_4^+$-$N\ m^{-2}\ yr^{-1}$) | 0.3 (0.0) ab A† | 0.5 (0.3) a | 0.2 (0.01) bc | 0.3 (0.0) ab | 0.2 (0.0) c |
| Nitrate ($g\ NO_3^-$-$N\ m^{-2}\ yr^{-1}$) | 0.1 (0.1) ab | 0.1 (0.1) ab A | 0.0 (0.0) b | 0.6 (0.3) a | 0.1 (0.0) ab |
| Dissolved organic N ($g\ N\ m^{-2}\ yr^{-1}$) | 0.2 (0.0) A† | 0.1 (0.0) | 0.1 (0.0) | 0.2 (0.1) | 0.1 (0.0) |
| Total dissolved N ($g\ N\ m^{-2}\ yr^{-1}$) | 0.6 (0.1) ab† A† | 0.8 (0.3) ab† | 0.4 (0.0) b† | 1.1 (0.3) a† | 0.4 (0.1) b† |
| Dissolved organic C ($g\ C\ m^{-2}\ yr^{-1}$) | 4.2 (0.5) bc | 6.2 (1.5) ab | 3.9 (0.2) c | 7.3 (0.2) a | 4.2 (0.4) bc |
| Sodium ($g\ Na\ m^{-2}\ yr^{-1}$) | 3.8 (0.4) b A | 3.7 (0.8) b | 3.1 (0.3) b | 13.1 (7.6) a | 3.1 (0.5) b |
| Potassium ($g\ K\ m^{-2}\ yr^{-1}$) | 0.4 (0.1) | 0.4 (0.2) | 0.4 (0.1) | 0.7 (0.2) | 0.4 (0.1) |
| Calcium ($g\ Ca\ m^{-2}\ yr^{-1}$) | 1.0 (0.1) b A | 1.2 (0.3) b | 0.9 (0.1) b | 4.6 (1.3) a | 1.0 (0.2) b |

| | | | | | |
|---|---|---|---|---|---|
| Magnesium ($g\ Mg\ m^{-2}\ yr^{-1}$) | 0.4 (0.0) b A | 0.4 (0.1) b | 0.4 (0.1) b | 0.9 (0.2) a | 0.3 (0.1) b |
| Total aluminum ($g\ Al\ m^{-2}\ yr^{-1}$) | 0.4 (0.1) b A | 0.3 (0.1) b | 0.4 (0.0) b | 2.3 (1.3) a | 0.2 (0.0) b |
| Total iron ($g\ Fe\ m^{-2}\ yr^{-1}$) | 0.20 (0.10) | 0.02 (0.01) | 0.03 (0.01) | 0.04 (0.00) | 0.10 (0.10) |
| Total manganese ($g\ Mn\ m^{-2}\ yr^{-1}$) | 0.02 (0.01) | 0.03 (0.02) | 0.01 (0.01) | 0.03 (0.00) | 0.01 (0.00) |
| Total phosphorus ($g\ P\ m^{-2}\ yr^{-1}$) | 0.01 (0.00) a† | 0.01 (0.00) abc† | 0.00 (0.00) c† | 0.01 (0.0) ab† | 0.01 (0.00) bc† |
| Total sulfur ($g\ S\ m^{-2}\ yr^{-1}$) | 0.20 (0.00) ab | 0.20 (0.10) ab | 0.13 (0.01) b | 0.24 (0.0) a | 0.15 (0.0) ab |
| Total silica ($g\ Si\ m^{-2}\ yr^{-1}$) | 0.7 (0.2) A† | 0.6 (0.3) | 0.4 (0.1) | 0.4 (0.1) | 0.3 (0.1) |
| Chloride ($g\ Cl\ m^{-2}\ yr^{-1}$) | 10.5 (0.9) b A | 11.5 (2.4) b | 9.1 (0.6) b | 38.0 (6.7) a | 7.8 (1.2) b |
| clay Acrisol soil | | | | | |
| Ammonium ($g\ NH_4^+\text{-}N\ m^{-2}\ yr^{-1}$) | 0.2 (0.0) B† | 0.2 (0.0) | 0.2 (0.0) | 0.2 (0.0) | 0.2 (0.0) |
| Nitrate ($g\ NO_3^-\text{-}N\ m^{-2}\ yr^{-1}$) | 0.1 (0.1) | 0.0 (0.0) B | 0.3 (0.2) | 1.1 (1.1) | 0.0 (0.0) |
| Dissolved organic N ($g\ N\ m^{-2}\ yr^{-1}$) | 0.1 (0.0) B† | 0.1 (0.0) | 0.1 (0.0) | 0.1 (0.0) | 0.1 (0.0) |
| Total dissolved N ($g\ N\ m^{-2}\ yr^{-1}$) | 0.3 (0.1) B† | 0.3 (0.0) | 0.6 (0.2) | 1.4 (1.1) | 0.3 (0.0) |

| | | | | | |
|---|---|---|---|---|---|
| Dissolved organic C ($g\ C\ m^{-2}\ yr^{-1}$) | 3.4 (0.4) c | 5.4 (0.7) ab | 3.6 (0.2) bc | 6.2 (1.4) a | 5.6 (1.0) ab |
| Sodium ($g\ Na\ m^{-2}\ yr^{-1}$) | 2.5 (0.4) b B | 3.2 (0.3) b | 2.5 (0.1) b | 6.3 (1.8) a | 3.3 (0.6) b |
| Potassium ($g\ K\ m^{-2}\ yr^{-1}$) | 0.3 (0.0) | 0.3 (0.1) | 0.3 (0.1) | 0.5 (0.1) | 0.2 (0.1) |
| Calcium ($g\ Ca\ m^{-2}\ yr^{-1}$) | 0.7 (0.1) B | 0.9 (0.0) | 0.8 (0.1) | 1.0 (0.2) | 0.7 (0.1) |
| Magnesium ($g\ Mg\ m^{-2}\ yr^{-1}$) | 0.2 (0.0) b B | 0.3 (0.0) b | 0.3 (0.0) b | 0.6 (0.1) a | 0.2 (0.1) b |
| Total aluminum ($g\ Al\ m^{-2}\ yr^{-1}$) | 0.2 (0.0) B | 0.2 (0.1) | 0.3 (0.1) | 0.3 (0.1) | 0.1 (0.0) |
| Total iron ($g\ Fe\ m^{-2}\ yr^{-1}$) | 0.02 (0.00) | 0.03 (0.00) | 0.02 (0.00) | 0.01 (0.0) | 0.06 (0.05) |
| Total manganese ($g\ Mn\ m^{-2}\ yr^{-1}$) | 0.01 (0.00) | 0.01 (0.00) | 0.01 (0.00) | 0.09 (0.07) | 0.02 (0.00) |
| Total phosphorus ($g\ P\ m^{-2}\ yr^{-1}$) | 0.01 (0.00) | 0.01 (0.00) | 0.01 (0.00) | 0.01 (0.00) | 0.02 (0.01) |
| Total sulfur ($g\ S\ m^{-2}\ yr^{-1}$) | 0.16 (0.0) ab | 0.15 (0.0) ab | 0.14 (0.0) b | 0.17 (0.0) a | 0.17 (0.0) ab |
| Total silica ($g\ Si\ m^{-2}\ yr^{-1}$) | 0.3 (0.1) b B† | 0.7 (0.1) ab | 0.3 (0.0) b | 1.3 (0.6) a | 0.8 (0.3) ab |
| Chloride ($g\ Cl\ m^{-2}\ yr^{-1}$) | 6.0 (0.3) B | 8.2 (1.3) | 6.9 (1.0) | 9.8 (3.0) | 5.6 (0.6) |

**Table 5.** Mean (± SE, n = 4, except for oil palm *n* = 3) nitrogen and base cation retention efficiency in soils under different land uses within two landscapes (loam and clay Acrisol soils) in Jambi, Sumatra, Indonesia. Mean followed by different lower case letters indicate significant differences among land uses within each landscape and different upper case letters indicate significant differences between landscapes for each reference land use (Linear mixed effects models with Fisher's LSD test at $P \leq 0.05$, and † at $P = 0.07$ for marginal significance).

| Characteristic | Forest | Jungle rubber | Rubber | Oil palm |
|---|---|---|---|---|
| | | loam Acrisol soil | | |
| N retention efficiency (mg N $m^{-2}$ $d^{-1}$/mg N $m^{-2}$ $d^{-1}$) | 0.997 (0.000) a B | 0.996 (0.001) a B† | 0.998 (0.000) a | 0.995 (0.001) b |
| Base cation retention efficiency ($mol_{charge}$ $m^{-2}$ $yr^{-1}$/ $mol_{charge}$ $m^{-2}$) | 0.455 (0.094) a† B | 0.591 (0.088) a† B† | 0.699 (0.08259) a† | 0.280 (0.128) b† |
| | | clay Acrisol soil | | |
| N retention efficiency (mg N $m^{-2}$ $d^{-1}$/mg N $m^{-2}$ $d^{-1}$) | 0.999 (0.000) A | 0.999 (0.000) A† | 0.997 (0.001) | 0.998 (0.001) |
| Base cation retention efficiency ($mol_{charge}$ $m^{-2}$ $yr^{-1}$/ $mol_{charge}$ $m^{-2}$) | 0.812 (0.084) A | 0.852 (0.083) A† | 0.841 (0.025) | 0.894 (0.028) |

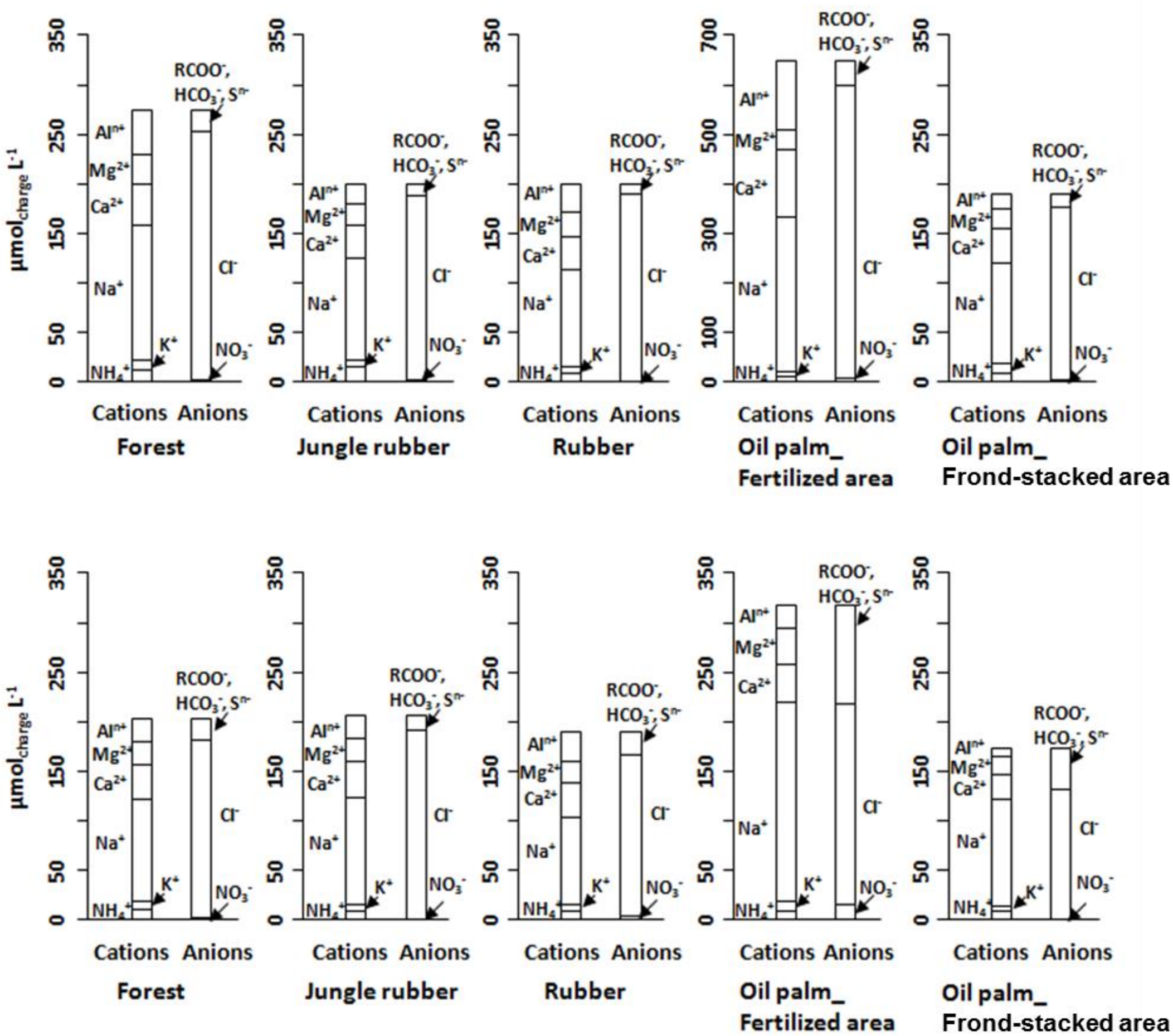

**Figure 1.** Partial cation-anion charge balance of the major solutes (with concentrations >0.03 mg $l^{-1}$) in soil water at a depth of 1.5 m in different land uses on the loam (top panel) and clay (bottom panel) Acrisol soils in Jambi, Sumatra, Indonesia. The y-axis scale of the oil palm fertilized area in the loam Acrisol soil is twice than the other land uses.

**Appendix A. Soil and vegetation characteristics, and Pearson correlations among solute concentrations in each land use within each landscape**

**Table A1.** Soil characteristics in the top 0.1 m of soil (except for clay content, which is for 1-2 m) in different land uses within two landscapes (loam and clay Acrisol soils) in Jambi, Sumatra, Indonesia. Mean (± SE, n = 4, except for clay content n = 3) followed by different lower case letters indicate significant differences among land uses within each landscape and different upper case letters indicate significant differences between landscapes for each reference land use (Linear mixed effects models with Fisher's LSD test at $P \leq 0.05$, and † at $P \leq 0.09$ for marginal significance). These soil characteristics were reported by Allen et al. (2015).

| Characteristic / land use | Forest | Jungle rubber | Rubber plantation | Oil palm plantation |
|---|---|---|---|---|
| | | loam Acrisol soil | | |
| Bulk density (g $cm^{-3}$) | 1.0 (0.04) ab | 0.9 (0.03) b A | 1.1 (0.1) a | 1.1 (0.1) a |
| pH (1:4 $H_2O$) | 4.3 (0.04) b† | 4.3 (0.03) b† B | 4.5 (0.1) ab† | 4.5 (0.1) a† |
| Soil organic C (kg C $m^{-2}$) | 2.6 (0.2) | 2.7 (0.3) B | 2.0 (0.3) | 1.8 (0.2) |
| Total N (g N $m^{-2}$) | 182.9 (10.8) | 186.1 (11.0) B | 172.6 (23.8) | 145.0 (13.5) |
| C:N ratio | 14.3 (0.2) a | 13.7 (0.8) a | 11.7 (0.7) b | 12.5 (0.5) ab |
| Effective cation exchange capacity (mmolc $kg^{-1}$) | 44.8 (5.0) | 40.6 (7.6) B | 46.0 (5.4) | 39.5 (7.9) |
| Base saturation (%) | 10.6 (0.5) b† B | 16.0 (2.2) ab† | 21.1 (7.5) ab† | 27.9 (5.4) a† |
| Potassium (g K $m^{-2}$) | 3.3 (0.3) | 2.6 (0.2) B | 3.4 (0.8) | 2.1 (0.8) |
| Sodium (g Na $m^{-2}$) | 0.5 (0.1) c B | 1.5 (0.2) b B | 1.4 (0.1) b | 3.9 (1.1) a |
| Calcium (g Ca $m^{-2}$) | 5.5 (2.0) | 6.9 (0.8) B† | 14.5 (7.1) | 18.5 (7.4) |
| Magnesium (g Mg $m^{-2}$) | 1.8 (0.1) | 2.0 (0.3) B | 3.4 (1.4) | 1.7 (0.9) |
| Aluminum (g Al $m^{-2}$) | 33.1 (3.5) | 29.6 (6.6) B | 30.7 (4.3) | 23.5 (2.7) |

| | | | | |
|---|---|---|---|---|
| Iron (g Fe m$^{-2}$) | 0.8 (0.1) a B | 0.3 (0.02) bc B | 0.3 (0.1) c | 0.5 (0.02) ab |
| Manganese (g Mn m$^{-2}$) | 0.3 (0.1) | 0.4 (0.2) B | 0.8 (0.3) | 0.5 (0.2) |
| Bray-extractable phosphorus (g P m$^{-2}$) | 0.5 (0.1) B | 0.7 (0.1) | 0.5 (0.1) | 0.8 (0.1) |
| Clay at 1.0-1.5 m (%) | 33.3 (7.6) | 42.4 (9.9) | 46.1 (9.9) | 43.3 (2.8) |
| Clay at 1.5-2.0 m (%) | 37.3 (8.7) | 44.5 (10.0) | 43.4 (6.5) | 47.6 (4.5) |
| clay Acrisol soil | | | | |
| Bulk density (g cm$^{-3}$) | 1.0 (0.1) | 0.8 (0.1) B | 0.9 (0.1) | 0.9 (0.1) |
| pH (1:4 $H_2O$) | 4.2 (0.04) b | 4.5 (0.04) a A | 4.5 (0.1) a | 4.4 (0.04) a |
| Soil organic C (kg C m$^{-2}$) | 3.3 (0.5) | 4.3 (0.4) A | 2.8 (0.4) | 3.5 (0.2) |
| Total N (g N m$^{-2}$) | 263.4 (67.1) | 331.4 (34.1) A | 198.9 (32.5) | 260.2 (22.6) |
| C:N ratio | 13.1 (1.3) | 13.0 (0.3) | 14.3 (0.6) | 13.5 (0.2) |
| Effective cation exchange capacity (mmolc kg$^{-1}$) | 94.3 (40.8) | 124.5 (25.5) A | 71.3 (22.3) | 78.1 (8.4) |
| Base saturation (%) | 22.9 (5.6) A | 23.2 (5.8) | 20.1 (2.6) | 37.5 (7.1) |
| Potassium (g K m$^{-2}$) | 9.4 (3.9) | 9.6 (2.6) A | 4.2 (1.1) | 4.8 (0.9) |
| Sodium (g Na m$^{-2}$) | 3.6 (0.8) A | 4.2 (0.2) A | 3.7 (1.3) | 1.9 (1.3) |
| Calcium (g Ca m$^{-2}$) | 32.3(21.2) | 33.3 (10.9) A† | 14.7 (2.8) | 59.1 (19.5) |
| Magnesium (g Mg m$^{-2}$) | 7.3 (3.9) | 12.0 (4.1) A | 4.0 (0.9) | 3.5 (0.8) |
| Aluminum (g Al m$^{-2}$) | 50.9 (22.7) | 76.6 (15.6) A | 47.2 (17.6) | 34.4 (2.0) |
| Iron (g Fe m$^{-2}$) | 3.7 (1.1) a A | 3.0 (0.4) a A | 2.3 (0.6) a | 0.7 (0.3) b |
| Manganese (g Mn m$^{-2}$) | 4.5 (3.1) | 2.5 (0.7) A | 1.5 (0.4) | 3.4 (1.3) |
| Bray-extractable phosphorus (g P m$^{-2}$) | 1.4 (0.1) ab A | 0.8 (0.1) bc | 0.4 (0.04) c | 4.7 (1.5) a |
| Clay at 1.0-1.5 m (%) | 39.0 (13.0) | 62.8 (12.6) | 40.8 (10.3) | 62.8 (3.7) |
| Clay at 1.5-2.0 m (%) | 41.3 (11.2) | 46.6 (16.2) | 36.5 (10.8) | 63.3 (6.1) |

**Table A2.** Mean (± SE, *n* = 4) tree density, diameter at breast height (DBH), basal area, height, cumulative fine root mass in the top 1 m depth and the most common tree species with DBH ≥ 0.10 m in different land uses within two landscapes (loam and clay Acrisol soils) in Jambi, Sumatra, Indonesia. The vegetation characteristics (e.g. tree density, DBH, basal area, and height) were reported by Kotowska et al. (2015), while the most common tree species with DBH ≥ 0.10 m were recorded based on trees found in five subplots (5 m x 5 m) of each replicate plot (50 m x 50 m) which had ≥ 20 individuals, except Fabaceae spp., which had < 20 individuals (reported by Rembold et al. (unpublished data)). The fine root mass in the top 1-m soil depth was measured in our present study. Mean of fine root mass followed by different lower case letters indicate significant differences among land uses within each landscape (Linear mixed effects models with Fisher's LSD test at $P \leq 0.05$, and † at $P \leq 0.09$ for marginal significance).

| Characteristics | Forest | Jungle rubber | Rubber | Oil palm |
|---|---|---|---|---|
| | loam Acrisol soil | | | |
| Plantation age (years) | not determined (ND) | ND | 14 – 17 | 12 – 16 |
| Tree density (trees ha$^{-1}$) | 658 (26) | 525 (60) | 440 (81) | 140 (4) |
| DBH (cm) | 21.0 (0.5) | 16.8 (0.5) | 17.8 (1.2) | not applicable (NA) |
| Basal area (m$^2$ ha$^{-1}$) | 30.7 (1.0) | 16.6 (0.4) | 12.2 (1.6) | NA |
| Tree height (m) | 20.0 (0.6) | 14.0 (0.2) | 13.4 (0.5) | 4.9 (0.6) |
| Fine root mass in the top 1-m soil depth (g m$^{-2}$) | 290.2 (82.6) $_{ab†}$ | 143.9 (33.0) $_b$ | 188.2 (37.6) $_b$ | 356.8 (49.9) $_a$ |

| | | | | |
|---|---|---|---|---|
| Common trees species | *Aporosa spp., Burseraceae spp., Dipterocarpaceae spp., Fabaceae spp., Gironniera spp., Myrtaceae spp., Plaquium spp., Porterandia sp., Shorea spp.* | *Alstonia spp., Artocarpus spp., Fabaceae sp., Hevea sp., Macaranga spp., Porterandia sp., Sloetia sp.* | *Hevea brasiliensis* | *Elaeis guineensis* |
| | | clay Acrisol soil | | |
| Plantation age (years) | ND | ND | 7 – 16 | 9 – 13 |
| Tree density (trees ha$^{-1}$) | 471 (31) | 685 (72) | 497 (15) | 134 (6) |
| DBH (cm) | 23.0 (0.4) | 17.3 (0.6) | 15.2 (0.7) | NA |
| Basal area (m$^2$ ha$^{-1}$) | 29.4 (1.7) | 21.1 (1.4) | 10.0 (1.4) | NA |
| Tree height (m) | 17.0 (0.5) | 15.2 (0.3) | 13.4 (0.1) | 4.0 (0.3) |
| Fine root mass in the top 1-m soil depth (g m$^{-2}$) | 140.4 (33.0) $_c$ | 402.2 (65.9) $_b$ | 309.6 (16.0) $_{bc}$ | 630.1 (86.2) $_a$ |
| Common tree species | *Archidendron sp., Baccaurea spp., Ochanostachys sp.* | *Artocarpus spp., Endospermum sp., Hevea sp., Macaranga spp.* | *Hevea brasiliensis* | *Elaeis guineensis* |

**Table A3.** Pearson correlations among element concentrations (mg $L^{-1}$) in soil solution (1.5-m depth) of the different land uses on the loam Acrisol soil in Jambi, Sumatra, Indonesia. Correlations were carried out using monthly averages of four replicate plots per land use (*n* = 12 monthly measurements in 2013). Elements that had concentrations < 0.03 mg $L^{-1}$ (total Fe, total Mn, and total P) and total Si (that did not show correlation with other elements) are not reported below.

| Element | $NH_4^+$-N | $NO_3^-$-N | DOC | $Na^+$ | $K^+$ | $Ca^{2+}$ | $Mg^{2+}$ | Total Al | Total S | $Cl^-$ |
|---|---|---|---|---|---|---|---|---|---|---|
| **Forest** | | | | | | | | | | |
| DON | 0.79 [c] | -0.24 | 0.77 [c] | 0.36 | 0.43 | 0.80 [c] | 0.77 [c] | 0.84 [c] | -0.17 | 0.86 [c] |
| $NH_4^+$-N | | 0.22 | 0.48 | 0.23 | 0.64 [b] | 0.67 [b] | 0.65 [b] | 0.58 [b] | 0.30 | 0.58 [b] |
| $NO_3^-$-N | | | -0.12 | -0.09 | 0.35 | -0.26 | -0.25 | -0.45 | 0.63 [b] | -0.47 |
| DOC | | | | 0.36 | 0.45 | 0.72 [c] | 0.71 [c] | 0.73 [c] | -0.02 | 0.68 [b] |
| $Na^+$ | | | | | 0.58 [b] | 0.53 [a] | 0.46 | 0.34 | 0.23 | 0.45 |
| $K^+$ | | | | | | 0.51 [a] | 0.45 | 0.29 | 0.71 [c] | 0.33 |
| $Ca^{2+}$ | | | | | | | 0.99 [c] | 0.94 [c] | 0.00 | 0.92 [c] |
| $Mg^{2+}$ | | | | | | | | 0.95 [c] | -0.03 | 0.92 [c] |
| Total Al | | | | | | | | | -0.28 | 0.95 [c] |
| Total S | | | | | | | | | | -0.23 |
| **Jungle rubber** | | | | | | | | | | |
| DON | 0.80 [c] | 0.28 | 0.77 [c] | 0.72 [c] | 0.85 [c] | 0.72 [c] | 0.79 [c] | 0.30 | 0.60 [b] | 0.68 [b] |
| $NH_4^+$-N | | 0.32 | 0.73 [c] | 0.35 | 0.77 [c] | 0.53 [a] | 0.67 [b] | 0.55 [b] | 0.17 | 0.79 [c] |
| $NO_3^-$-N | | | 0.35 | 0.17 | 0.20 | 0.65 [b] | 0.62 [b] | 0.61 [b] | -0.11 | 0.65 [b] |
| DOC | | | | 0.63 [b] | 0.76 [c] | 0.51 [a] | 0.53 [a] | 0.13 | 0.57 [b] | 0.49 [a] |
| $Na^+$ | | | | | 0.80 [c] | 0.58 [b] | 0.55 [b] | -0.18 | 0.93 [c] | 0.29 |

| | | | | | | | | | | |
|---|---|---|---|---|---|---|---|---|---|---|
| $K^+$ | | | | | | $0.65^{b}$ | $0.70^{c}$ | 0.12 | $0.65^{b}$ | $0.60^{b}$ |
| $Ca^{2+}$ | | | | | | | $0.97^{c}$ | $0.56^{b}$ | 0.32 | $0.84^{c}$ |
| $Mg^{2+}$ | | | | | | | | $0.65^{b}$ | 0.27 | $0.93^{c}$ |
| Total Al | | | | | | | | | -0.47 | $0.85^{c}$ |
| Total S | | | | | | | | | | -0.02 |
| | | | | | **Rubber** | | | | | |
| DON | -0.12 | -0.32 | $0.53^{a}$ | 0.04 | $0.65^{b}$ | 0.37 | 0.65 | $0.67^{b}$ | -0.28 | 0.39 |
| $NH_4^+$-N | | 0.10 | 0.31 | $0.61^{b}$ | -0.05 | 0.17 | -0.07 | -0.41 | $0.65^{b}$ | -0.18 |
| $NO_3^-$-N | | | -0.25 | 0.25 | -0.48 | 0.42 | 0.15 | -0.09 | 0.26 | 0.31 |
| DOC | | | | $0.50^{a}$ | 0.46 | $0.51^{a}$ | $0.50^{a}$ | 0.29 | 0.30 | 0.34 |
| $Na^+$ | | | | | 0.17 | 0.46 | 0.08 | -0.34 | $0.85^{c}$ | 0.00 |
| $K^+$ | | | | | | 0.24 | $0.55^{b}$ | $0.54^{a}$ | -0.15 | 0.38 |
| $Ca^{2+}$ | | | | | | | $0.81^{c}$ | 0.40 | 0.27 | $0.72^{c}$ |
| $Mg^{2+}$ | | | | | | | | $0.84^{c}$ | -0.26 | $0.92^{c}$ |
| Total Al | | | | | | | | | $-0.70^{c}$ | $0.83^{c}$ |
| Total S | | | | | | | | | | -0.35 |
| | | | | | **Oil palm fertilized areas** | | | | | |
| DON | -0.28 | 0.08 | -0.18 | $-0.57^{b}$ | -0.12 | 0.16 | 0.31 | 0.50 | -0.06 | 0.08 |
| $NH_4^+$-N | | $0.54^{a}$ | -0.12 | 0.00 | 0.50 | 0.15 | 0.37 | 0.46 | 0.22 | 0.46 |
| $NO_3^-$-N | | | -0.12 | 0.14 | -0.02 | -0.49 | 0.00 | $0.63^{b}$ | -0.38 | 0.10 |
| DOC | | | | -0.22 | 0.08 | 0.02 | 0.29 | -0.17 | 0.40 | -0.47 |
| $Na^+$ | | | | | -0.12 | -0.45 | -0.45 | -0.37 | -0.38 | 0.22 |
| $K^+$ | | | | | | $0.58^{b}$ | 0.43 | -0.17 | $0.58^{b}$ | 0.27 |

| | | | | | | | | | | |
|---|---|---|---|---|---|---|---|---|---|---|
| $Ca^{2+}$ | | | | | | | 0.48 | -0.19 | 0.79 $^{c}$ | 0.45 |
| $Mg^{2+}$ | | | | | | | | 0.40 | 0.72 $^{c}$ | 0.41 |
| Total Al | | | | | | | | | -0.16 | 0.27 |
| Total S | | | | | | | | | | 0.30 |
| **Oil palm frond-stacked areas** | | | | | | | | | | |
| DON | -0.38 | 0.38 | 0.22 | -0.38 | 0.24 | -0.47 | -0.16 | 0.47 | -0.59 $^{b}$ | 0.04 |
| $NH_4^+$-N | | 0.07 | 0.23 | 0.40 | 0.25 | 0.04 | 0.08 | -0.17 | 0.42 | 0.06 |
| $NO_3^-$-N | | | 0.61 $^{b}$ | 0.12 | 0.56 $^{b}$ | -0.26 | -0.21 | 0.11 | 0.20 | 0.02 |
| DOC | | | | -0.10 | 0.57 $^{b}$ | -0.38 | -0.55 $^{b}$ | -0.28 | 0.22 | -0.42 |
| $Na^+$ | | | | | 0.09 | 0.23 | 0.22 | -0.35 | 0.61 $^{b}$ | 0.09 |
| $K^+$ | | | | | | -0.27 | -0.21 | -0.07 | 0.29 | -0.06 |
| $Ca^{2+}$ | | | | | | | 0.83 $^{c}$ | 0.30 | -0.15 | 0.72 $^{c}$ |
| $Mg^{2+}$ | | | | | | | | 0.63 $^{b}$ | -0.41 | 0.95 $^{c}$ |
| Total Al | | | | | | | | | -0.81 $^{c}$ | 0.79 $^{c}$ |
| Total S | | | | | | | | | | -0.48 |

$^{a}P \leq 0.09$, $^{b}P \leq 0.05$, $^{c}P \leq 0.01$.

Biogeosciences Discussions Open Access EGU

**Table A4.** Pearson correlations among element concentrations (mg $L^{-1}$) in soil solution (1.5-m depth) of the different land uses on the clay Acrisol soil in Jambi, Sumatra, Indonesia. Correlations were carried out using monthly averages of four replicate plots per land use (*n* = 12 monthly measurements in 2013). Element that had concentrations < 0.03 mg $L^{-1}$ (total Fe, total Mn, and total P) and total Si (that did not show correlation with other elements) are not reported below.

| Element | $NH_4^+$-N | $NO_3^-$-N | DOC | $Na^+$ | $K^+$ | $Ca^{2+}$ | $Mg^{2+}$ | Total Al | Total S | $Cl^-$ |
|---|---|---|---|---|---|---|---|---|---|---|
| | | | | | **Forest** | | | | | |
| DON | 0.10 | -0.39 | 0.57 $^{b}$ | 0.32 | 0.53 $^{a}$ | 0.17 | 0.20 | -0.28 | 0.25 | -0.20 |
| $NH_4^+$-N | | -0.48 | 0.81 $^{c}$ | 0.63 $^{b}$ | 0.23 | 0.51 $^{a}$ | 0.28 | -0.11 | -0.27 | 0.09 |
| $NO_3^-$-N | | | -0.48 | -0.24 | -0.18 | -0.05 | -0.03 | 0.36 | 0.12 | 0.37 |
| DOC | | | | 0.66 $^{b}$ | 0.41 | 0.48 | 0.31 | -0.25 | -0.15 | -0.06 |
| $Na^+$ | | | | | 0.69 $^{b}$ | 0.52 $^{a}$ | 0.54 $^{a}$ | -0.22 | -0.24 | -0.10 |
| $K^+$ | | | | | | 0.74 $^{c}$ | 0.88 $^{c}$ | 0.22 | -0.17 | 0.26 |
| $Ca^{2+}$ | | | | | | | 0.93 $^{c}$ | 0.54 $^{a}$ | -0.29 | 0.70 $^{c}$ |
| $Mg^{2+}$ | | | | | | | | 0.52 $^{a}$ | -0.34 | 0.59 $^{b}$ |
| Total Al | | | | | | | | | -0.15 | 0.94 $^{c}$ |
| Total S | | | | | | | | | | -0.10 |
| | | | | | **Jungle rubber** | | | | | |
| DON | 0.23 | 0.55 $^{b}$ | 0.58 $^{b}$ | 0.19 | 0.69 $^{c}$ | 0.50 $^{a}$ | 0.63 $^{b}$ | 0.70 $^{c}$ | -0.22 | 0.49 $^{a}$ |
| $NH_4^+$-N | | 0.01 | 0.36 | 0.35 | 0.35 | 0.29 | 0.29 | 0.16 | 0.31 | 0.18 |
| $NO_3^-$-N | | | 0.32 | 0.30 | 0.49 $^{a}$ | 0.51 $^{a}$ | 0.50 $^{a}$ | 0.35 | 0.13 | 0.42 |
| DOC | | | | -0.24 | 0.11 | -0.14 | -0.05 | 0.29 | 0.06 | -0.20 |
| $Na^+$ | | | | | 0.68 $^{c}$ | 0.84 $^{c}$ | 0.73 $^{c}$ | 0.01 | 0.52 $^{a}$ | 0.66 $^{b}$ |

| | | | | | | | | | | |
|---|---|---|---|---|---|---|---|---|---|---|
| $K^+$ | | | | | | $0.87^{c}$ | $0.93^{c}$ | $0.63^{b}$ | 0.09 | $0.84^{c}$ |
| $Ca^{2+}$ | | | | | | | $0.97^{c}$ | $0.50^{a}$ | 0.09 | $0.95^{c}$ |
| $Mg^{2+}$ | | | | | | | | $0.66^{b}$ | -0.04 | $0.97^{c}$ |
| Total Al | | | | | | | | | $-0.62^{b}$ | $0.68^{b}$ |
| Total S | | | | | | | | | | -0.18 |
| **Rubber** | | | | | | | | | | |
| DON | -0.20 | -0.18 | 0.21 | -0.29 | 0.41 | 0.40 | $0.55^{b}$ | $0.65^{b}$ | $-0.57^{b}$ | 0.48 |
| $NH_4^+$-N | | 0.22 | $0.81^{c}$ | $0.85^{c}$ | 0.47 | 0.19 | 0.10 | -0.20 | $0.52^{a}$ | -0.06 |
| $NO_3^-$-N | | | -0.07 | -0.16 | -0.44 | $-0.68^{b}$ | $-0.60^{b}$ | -0.38 | 0.05 | $-0.63^{b}$ |
| DOC | | | | $0.79^{c}$ | $0.71^{c}$ | $0.54^{a}$ | 0.45 | 0.20 | 0.43 | 0.30 |
| $Na^+$ | | | | | $0.61^{b}$ | 0.38 | 0.21 | -0.15 | $0.65^{b}$ | 0.07 |
| $K^+$ | | | | | | $0.67^{b}$ | $0.66^{b}$ | 0.46 | 0.08 | $0.64^{b}$ |
| $Ca^{2+}$ | | | | | | | $0.93^{c}$ | $0.73^{c}$ | -0.16 | $0.83^{c}$ |
| $Mg^{2+}$ | | | | | | | | $0.88^{c}$ | -0.39 | $0.93^{c}$ |
| Total Al | | | | | | | | | $-0.58^{b}$ | $0.89^{c}$ |
| Total S | | | | | | | | | | -0.40 |
| **Oil palm fertilized areas** | | | | | | | | | | |
| DON | 0.02 | -0.09 | 0.49 | $0.70^{b}$ | $0.69^{b}$ | $0.67^{b}$ | 0.42 | 0.45 | $0.54^{a}$ | $0.63^{b}$ |
| $NH_4^+$-N | | 0.08 | 0.15 | 0.39 | 0.37 | 0.16 | 0.06 | 0.06 | 0.46 | -0.01 |
| $NO_3^-$-N | | | -0.18 | 0.03 | 0.46 | $0.51^{a}$ | -0.01 | 0.19 | 0.33 | -0.49 |
| DOC | | | | $0.52^{a}$ | $0.66^{b}$ | $0.56^{a}$ | 0.50 | $0.56^{a}$ | 0.25 | $0.70^{b}$ |
| $Na^+$ | | | | | $0.61^{b}$ | $0.61^{b}$ | 0.29 | 0.21 | $0.75^{c}$ | $0.55^{a}$ |
| $K^+$ | | | | | | $0.85^{c}$ | $0.74^{c}$ | $0.78^{c}$ | $0.52^{a}$ | $0.59^{b}$ |

| | | | | | | | | | | |
|---|---|---|---|---|---|---|---|---|---|---|
| $Ca^{2+}$ | | | | | | | 0.81 [c] | 0.74 [c] | 0.69 [b] | 0.64 [b] |
| $Mg^{2+}$ | | | | | | | | 0.95 [c] | 0.26 | 0.74 [c] |
| Total Al | | | | | | | | | 0.15 | 0.75 [c] |
| Total S | | | | | | | | | | 0.26 |
| **Oil palm frond-stacked areas** | | | | | | | | | | |
| DON | 0.19 | 0.34 | 0.15 | 0.49 [a] | 0.47 | 0.51 [a] | 0.23 | 0.29 | 0.28 | 0.36 |
| $NH_4^+$-N | | -0.07 | 0.27 | 0.21 | 0.38 | 0.11 | 0.06 | 0.07 | 0.13 | 0.09 |
| $NO_3^-$-N | | | -0.28 | 0.24 | 0.32 | 0.13 | -0.13 | 0.09 | 0.56 [b] | -0.05 |
| DOC | | | | 0.09 | 0.23 | 0.25 | 0.45 | 0.02 | -0.46 | 0.19 |
| $Na^+$ | | | | | 0.91 [c] | 0.94 [c] | 0.76 [c] | 0.91 [c] | 0.33 | 0.89 [c] |
| $K^+$ | | | | | | 0.88 [c] | 0.74 [c] | 0.80 [c] | 0.21 | 0.79 [c] |
| $Ca^{2+}$ | | | | | | | 0.90 [c] | 0.91 [c] | 0.10 | 0.95 [c] |
| $Mg^{2+}$ | | | | | | | | 0.81 [c] | -0.28 | 0.93 [c] |
| Total Al | | | | | | | | | 0.16 | 0.92 [c] |
| Total S | | | | | | | | | | -0.06 |

[a] $P \leq 0.09$, [b] $P \leq 0.05$, [c] $P \leq 0.01$.

**Appendix B. Fine root biomass and soil water model validation**

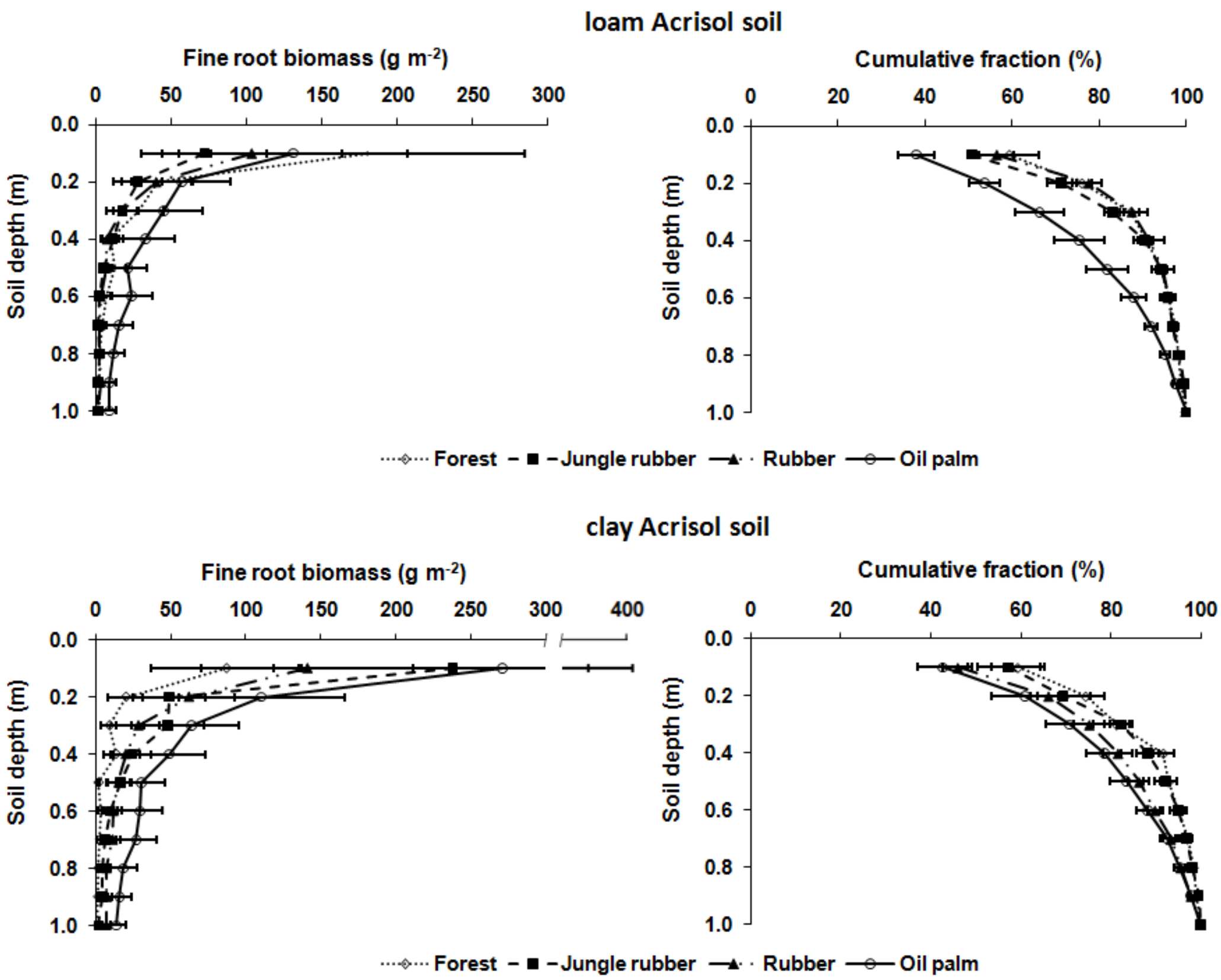

**Figure B1.** Fine root biomass (g m$^{-2}$) and distribution (%) down to a depth of 1 m in different
land uses within two landscapes (loam and clay Acrisol soils) in Jambi, Sumatra, Indonesia.
The root measurement was conducted in each replicate plot by digging a pit (1 m x 1.5 m x 2-
839 m depth) at about 2.5-m distance from an oil palm or a tree with a diameter at breast height of
840 ≥ 10 cm. Root mass were sampled using a metal block (20 cm x 20 cm x 10 cm) at 10-cm depth
interval from the top down to 1 m. Roots were carefully separated from the soil by washing
over a 2-mm mesh screen and the fine roots were collected in a basin placed underneath the
mesh screen. The roots were categorized into fine roots (≤ 2 mm diameter) and coarse roots (>2
844 mm diameter), dried in an oven at 70 $^0$C for 5 days and weighed.

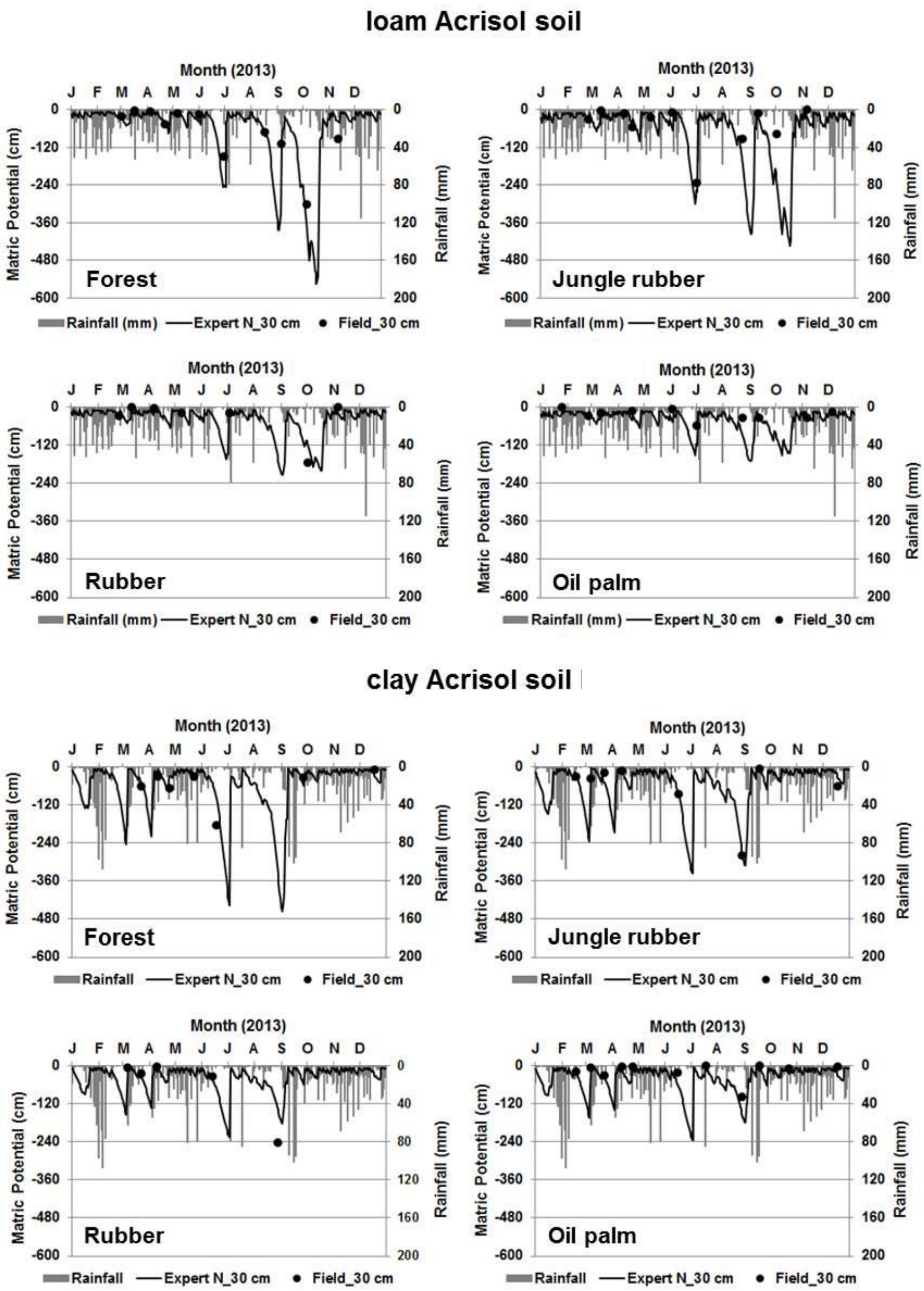

**Figure B2.** Validation between Expert N-modelled and field-measured matric potential at a depth of 0.3 m in different land uses within two landscapes (loam and clay Acrisol soils) in Jambi, Sumatra, Indonesia**.**