# Peer review of "Conversion of tropical forests to smallholder rubber and oil palm plantations impacts nutrient leaching losses and nutrient retention efficiency in highly weathered soils"

_Biogeosciences, 2018_

## Referee Comment (RC1) · Y. A. Teh (Referee) · 5 Jun 2018

GENERAL COMMENTS This is an interesting and very topically-relevant paper, given the large, global extent of oil palm production and the current drive to understand not only oil palm's environmental impacts, but also to derive potential mitigation options for small growers and large agribusinesses. The contrast between different soil types (e.g. loam versus clay) is also very instructive, given that oil palm is cultivated on a mixture of soil types all over the tropics, and capturing this range of variability will help stakeholders develop a better predictive understanding, that includes knowledge

of how soils properties (e.g. texture, etc.) play a role in modulating aqueous fluxes of nutrients. Lastly, the focus on smallholdings is also welcome, given that these systems form such a large part of the production landscape, and are often under-represented in existing projects which have focused on larger-scale industrial plantations.

Overall it is my assessment that this paper was clearly written and well-structured. The methods appeared wholly appropriate for the research questions and hypotheses tested here. The approach to data analysis and interpretation appears logical and well-reasoned. I therefore do not find that this paper requires too much modification prior to publication, since this is – in my view – a solid and rigorous piece of research, that will make a meaningful contribution to our wider understanding of the aqueous biogeochemistry of managed tropical landscapes in Southeast Asia.

However, I did have a few general remarks and suggestions for improvement. More specific comments are provided in the section which follows this one. First, I think it may be worthwhile re-organizing the information in the discussion around the major findings, listing the top-level or most important findings first. The current structure of the discussion generally follows the order in which the results are reported, but there could be some value in arranging information according to the most ground-breaking or high impact results, in order to maximise the impact of the most important findings on the reader. Structuring a discussion in this way can be especially effective for data-rich papers like this one, because the discussion sections for data-rich papers can sometimes become quite large and extensive, and it is possible for key messages to get lost due to the volume of information covered.

Second, another topic that is theoretically interesting and also policy-relevant is whether or not the investigators believe that over-fertilization is occurring for the rubber and oil palm systems? To phrase this another way, are the higher nutrient losses for rubber and oil palm because fertilizer inputs exceed plant/ecosystem demand, or because of the transport-reaction properties of the different soil types (e.g. do the exchange properties and rate of physical transport through the soil mean that the soil

exchange complex cannot retain some of the added nutrients)? If the answer is the former, then this suggests that growers could be reducing their inputs of some elements. If the answer is the latter, then mitigation options become more complex, because they may require new means of introducing fertilizers to the soil (e.g. slow release fertilizers, organic fertilizers, soil conditioners to enhance CEC, etc.). It would be interesting if the investigators could expand upon this topic further in the discussion.

Third, two aqueous fluxes not included in this study are throughfall and stemflow. This observation is not meant as a criticism per se, as I fully recognize that this was very comprehensive and in-depth study, and resources are always limited for large-scale field experiments like this one. However, it would be useful if the investigators could comment on whether they think that differences in throughfall and stemflow among the different land-uses could have resulted in differences in nutrient dynamics and loss? Throughfall and stemflow are potentially influenced by factors such as vegetation structure (e.g. plant density), leaf area and tissue chemistry, so it is possible that the different cover types (with different vegetation structure and properties) could have different patterns in throughfall and stemflow, with knock-on effects for soil nutrient dynamics.

SPECIFIC COMMENTS 1. Lines 49-51: Provide information for wider context: It is worthwhile emphasizing here that smallholdings are very common through SE Asia, and account for approximately 40 % of the land under production throughout the region. Therefore, while the smallholdings in Jambi may represent a larger proportion of land area than elsewhere in SE Asia, smallholdings are common and thus important to understand.

2. Line 147-150: Consider re-phrasing the description of the fertilization rates, as the current wording makes it a bit more difficult to understand. One option may be to break-up this sentence into two shorter sentences; one referring to the clay Acrisol and the other to the loam Acrisol.

3. Line 161: Minor question or point for clarification: Do the authors have any insight as

to where nutrient-acquiring roots proliferate in this system? Is it possible that sampling 1.3-1.5 m from the palm could slightly overestimate the rate of leaching loss? Oil palms tend to show the highest density of roots within 1 m of the plant stem; therefore, it is possible that by sampling outside of this region the investigators may underestimate plant uptake or overestimate leaching. Arguably, however, it is not clear if all the roots within 1 m of the palm stem are active or specialized for nutrient uptake, i.e. many of these roots may be dead or not directly involved in nutrient acquisition. Moreover, if the growers' practice is to apply fertilizer 1.3-1.5 m from the stem, then it is likely that this sampling scheme is likely to best represent actual trends in leaching. It is also possible that the roots produced 1.3-1.5 m from the stem are tracking nutrient availability and are specialized for nutrient uptake.

4. Line 163: Did the growers plant any understory plants for erosion control? If so, did the authors sample from these areas too? Although the biomass and uptake capacity of these herbaceous plants is likely to be low relative to mature palms, leaching patterns are likely to be different from unvegetated areas.

5. Lines 268-279: Do the investigators have an estimate for the nutrient input from throughfall and stemflow? If these data do not exist, is it possible to constrain these values in the model from similar systems? While rain water provides a useful end-member with which to estimate the nutrient content of "external" moisture inputs, it is possible that dry deposition of nutrients and leaching from aboveground plant parts could contribute to the nutrient input to soil. Especially if this region is near local sources of N pollution, it is possible that throughflow/stemflow could make a contribution to the overall N load to the soil.

6. Lines 317-320: What are the comparable values for ET, run-off and drainage for rubber and oil palm systems?

7. Lines 395-534: Given that the authors introduce testable hypotheses in the introduction, I think it's important to "close the circle" by referencing these hypotheses in the

discussion, and confirming if the authors' findings supported or falsified their hypotheses.

8.  Line 441: Further clarification required re: the phrase "higher rates of soil NH4+ cycling." For those who have not yet read Allen et al. (2015), does this phrase mean that the rate of NH4+ mineralization is greater, gross production and uptake of NH4+ is greater, or that the overall turnover of NH4+ is greater?

9.  Lines 491-534: One question and one comment: first, given the finding that fertilization is enhancing leaching losses in these smallholder landscapes, do the authors believe that the growers are over-fertilizing? Is the high rate of leaching loss because the plant demand is lower than nutrient supply, or is it because transport factors mean that the nutrients are lost before plants are able to take-up the nutrients? The authors expert assessment directly influences policy and management decisions; if it is an over-fertilization situation (i.e. plant demand « nutrient input), then the mitigation option would be to reduce fertilizer inputs. If it is an issue of transport (e.g. ion exchange sites are saturated or movement of soil solution is too rapid for efficient plant uptake), the different mitigation options suggest themselves (e.g. use of slower release fertilizers, or other technologies to reduce nutrient transport through the soil column). The conclusion that soil texture was the dominant influence (lines 528-529) tends to imply that the authors believe the second option is more likely (i.e. rapid transport leads to loss, rather than plant demand « nutrient input); however, it would be useful to hear the authors thoughts on this topic given its wider importance for mitigation of nutrient pollution.

My second point is a comment rather than a question. One of the challenges in predicting the behaviour of smallholder systems is that there is potentially a wider diversity of practices and fertilization schemes compared to large-scale industrial plantations. For instance, depending on the relative wealth or resources of individual growers, they may have better access to fertilizers than less fortunate growers. While this does not necessarily take away from the message that the authors are trying to convey here (i.e. that

certain types of more "intensive" or "invasive" land-use can show enhanced leaching losses), I think it is useful to discuss this source of potential variance and uncertainty, since it means that we have to develop better process-based models so that we can adequately predict flux from smallholder systems.

10. Table and figure legends: Minor pedantic point: throughout the table and figure legends, the authors refer to the loam Acrisol and clay Acrisol as two different "landscapes." While I do not consider this as problematic as such, I wonder if the phrase "soil orders" or "soil types" may be more intuitive for the reader, given that the reference for these two types of environments are the names of the soil orders?

Yit Arn Teh, University of Aberdeen, School of Biological Sciences, Aberdeen, AB24 3UU, Scotland, UK

---

## Referee Comment (RC2) · K. Fujii (Referee) · 13 Jun 2018

General comments: The paper has dealt with effects of land use change and soil texture on nutrient losses from the systems. The data are based on the proper methodology and the data are reasonable, but there is some room to add discussion before reaching conclusion.

One of major issues is soil classification (Acrisols). Sumatra soils are more or less affected by volcanic ash deposition. Soils are relatively young among Indonesian soils.

[Figure]

I am afraid whether the soils studied satisfy Acrisols' low clay activity. Please confirm the soil profile data especially in the Bt horizon. Low CEC/clay is required. In addition, both loam and clay Acrisols contain high contents of clays. Please clarify how two types were separated.

L447 The authors link between N leaching and the acid-buffering capacity of the soils, but link between N leaching and clay contents will be precise. Exchangeable Al as well as pH is a record of soil acidification, but quantitative link between N loss and soil ANC can not be supported by calculating proton budgets in soil.

There were some draughts or dry-wet cycles in Indonesia. This has strong impacts on solute concentration and leaching flux. I recommend to add correlation analyses between water flux (or soil water content) and solute concentration to check dilution or condensation effects by dry-wet cycles. This effect can affect annual nutrient loss as well. At least, adding discussion will improve manuscript.

Throughout the paper, the authors use the ambiguous term "soil fertility". The definition of soil fertility is not same among the readers. Please define it in the beginning of the paper. Most of soil scientists avoid to use the term "soil fertility" in scientific paper.

Specific comments: The authors regarded jungle rubber as original vegetation, but it is introduced from Brazil some hundreds of years ago. It is not native vegetation.

The authors ascribed the greater nutrient losses from loam Acrisols than those from clay Acrisols. However, tree composition is not same between two sites. The authors need to add careful discussion on this topic.

L525-527 erosion and enhanced microbial mineralization of the native SOM can also contribute to low SOC stocks in oil palm plantation.

L505-506 What data can support this statement?

Table A2 sp. or spp. should not be written in italic. Dipterocarpaceae spp. include Shorea spp. The tree composition should be re-checked.

Throughout the paper, "l-1" and "L-1" are used inconsistently. Please use term s consistently.

---

## Author Comment (AC2) · 19 Jul 2018

Author's response:

First, we extend our sincere thanks to Dr. Fujii for his insightful suggestions and comments that help improve our manuscript greatly. We describe how we have addressed his comments in our answers below.

1. One of major issues is soil classification (Acrisols). Sumatra soils are more or less affected by volcanic ash deposition. Soils are relatively young among Indonesian soils.

[Figure]

I am afraid whether the soils studied satisfy Acrisols' low clay activity. Please confirm the soil profile data especially in the Bt horizon. Low CEC/clay is required. In addition, both loam and clay Acrisols contain high contents of clays. Please clarify how two types were separated.

Author's response:

Soil particle size distribution was determined from the three sites (or replicate plots) of each land use within each landscape (which was subsequently classified according to the major soil texture group). In each site, soil samples for particle size analysis was taken from 6 depth intervals within 2-m depth. For the general soil texture classification, we took the values of depth-weighted average for each site, and then the averaged for the 12 sites (4 land uses x 3 sites) for each landscape. Similarly, cation exchange capacity (CEC) was determined from 2 to 5 samples per depth for each site. We also did an oxalate-extraction for Fe and Al, and these characteristics did not satisfy for an Andic property.

Our soil texture classification is based on the averages of the plots in order to come up with a general category of soil texture. The clay area had an average across sites of 48% clay, 27% silt and 25% sand. The loam area had particle size fractions that bordered between loam and (sandy) clay loam, and for ease in writing the classification in the manuscript, we generally termed this as loam (22-32% clay, 25-30% silt and 45-50% sand).

In the clay soils (with more than 40% clay), >8% increase in clay was observed in the subsoil. In the loam soils, the ratio of subsoil % clay to overlying layer was >1.2. Both of these metrics satisfied the criterion for an Argic horizon. These Argic horizons of our sites has CEC of < 24 cmolc kg-1 clay and base saturation of <10% (all reported in our earlier publication, Allen et al. 2016).

Author's changes in the manuscript:

As these characteristics were already reported in our earlier publication (Allen et al. 2016), we did not elaborate these in the present manuscript. However, in order to address this concern of Dr. Fujii, we inserted in section 2.1, after a short description of the loam and clay Acrisol soils, the following:

Detailed soil characteristics of these classifications are reported by Allen et al. (2016).

2. L447 The authors link between N leaching and the acid-buffering capacity of the soils, but link between N leaching and clay contents will be precise. Exchangeable Al as well as pH is a record of soil acidification, but quantitative link between N loss and soil ANC cannot be supported by calculating proton budgets in soil.

Author's response:

We agree with the reviewer's comments. As stated in the Results (L366-376), N leaching fluxes (and N and base cation retention efficiency) were significantly correlated with soil base saturation, ECEC and organic C which were, in turn, correlated with clay content. Thus, this L447 is a leap in our interpretation, and we changed this L447 (as a topic sentence for this Discussion on correlations of nutrient leaching fluxes with soil biochemical characteristics) to refer only to what were clearly reflected by the correlation tests.

Author's changes in the manuscript: We changed L447 to:

The influenced of soil texture on soil biochemical characteristics also linked to the leaching losses or, conversely, nutrient retention efficiency.

3. There were some droughts or dry-wet cycles in Indonesia. This has strong impacts on solute concentration and leaching flux. I recommend to add correlation analyses between water flux (or soil water content) and solute concentration to check dilution or condensation effects by dry-wet cycles. This effect can affect annual nutrient loss as well. At least, adding discussion will improve manuscript.

Author's response:
From the start of our data analyses, we have explored all correlation tests, including correlations of element concentrations with the modelled soil moisture content at 1.5-m depth (lysimeter sampling) in order to check for dilution or condensation; the correlation tests with soil moisture contents were conducted in a similar manner as those synthesized in Appendix Tables 3 and 4. The only significant correlation coefficients with soil moisture contents were found for the fertilized area of the oil palm plantations in the loam Acrisol soil for K, Ca, Mg and total S concentrations (r = -0.59 – -0.72, P ≤ 0.05, n = 12 monthly measurements for one year). However, the influence of soil water content on nutrient concentrations were anyway incorporated in the calculation of leaching fluxes (nutrient concentration x drainage flux), as stated in L244-249. Also, all the correlations that used the nutrient concentrations (Appendix Tables 3 and 4) were interpreted only to assess which cations were correlated with which anions (i.e., L334-337, L344-347, L358-361) in order to support the partial ionic charge balance of solutes, as depicted in Fig. 1.

Author's changes in the manuscript:

Considering that (1) these above correlation tests were only significant in four elements at one spatial category (fertilized area of oil palm in the loam soils) and (2) the influence of soil water on element concentrations were incorporated in the calculation of leaching fluxes, we will not add this in the Discussion. This is so that the main highlights of our findings will not be buried, as suggested by Dr. Teh's (reviewer 1) first major suggestion.

4. Throughout the paper, the authors use the ambiguous term "soil fertility". The definition of soil fertility is not same among the readers. Please define it in the beginning of the paper. Most of soil scientists avoid to use the term "soil fertility" in scientific paper.

Author's response:

We agree and take this suggestion. When we used the word soil fertility, we specified in brackets the soil biochemical characteristics that we used as basis. These were already done in the original manuscript version of the Introduction and M&M.

Introduction L61-62 - Soil texture affects nutrient leaching through its control on soil fertility (e.g., cation exchange capacity, decomposition, and nutrient cycling) and soil water-holding capacity.

M & M L126-128 - In summary, the soil textural difference leads to inherent differences in soil fertility (e.g., higher effective cation exchange capacity, base saturation, Bray-extractable P and lower Al saturation) in the clay than the loam Acrisols under forest and jungle rubber (Appendix Table A1).

Author's changes in the manuscript:

We specified the soil biochemical properties when we used the word soil fertility. Namely: Discussion, section 4.2, the last 2 sentences of 1st paragraph:

The lower annual nutrient leaching fluxes in clay as compared to loam Acrisols (i.e., TDN, Na, Ca, Mg; Table 4) were paralleled by higher gross rates of $NH_4^+$ production and immobilization (Allen et al., 2015), soil N stocks, ECEC, base saturation (Appendix Table A1) and water-holding capacity (Hassler et al., 2015). Our findings showed that soil texture regulated nutrient leaching losses and soil fertility (e.g., nutrient stocks and N-cycling rates) in these highly weathered Acrisol soils.

In the Abstract and Discussion, section 4.3 last sentence of 1st paragraph – we replaced 'soil fertility' with 'nutrient availability':

Our results showed that disruption of nutrient cycling between the soil and vegetation brought about by land-use conversion to rubber plantations, combined with the absence of soil amendments, had decreased nutrient leaching (Tables 3 and 4) as well soil nutrient availability (i.e., P stocks, microbial N, gross N mineralization rates; Allen et al., 2015; Allen et al., 2016).

In the Conclusion, we replaced soil fertility with soil nutrients or nutrient levels.

Specific comments:

1. The authors regarded jungle rubber as original vegetation, but it is introduced from Brazil some hundreds of years ago. It is not native vegetation.

Author's response:

We replaced the word 'original' with 'previous' (in the Introduction) or 'reference land use' (in the Discussion), which was what we actually meant. We use 'previous' or 'reference land use' to denote the land use immediately before the conversion to rubber and oil palm.

2. The authors ascribed the greater nutrient losses from loam Acrisols than those from clay Acrisols. However, tree composition is not same between two sites. The authors need to add careful discussion on this topic.

Author's response:

This is a very good point and we take this into consideration. Our interpretation that soil texture was the main factor influencing leaching losses is based on the following: a) total net primary production (aboveground + belowground), as an indicator of plant usage of soil nutrients, of the forest and jungle rubber did not differ between the loam and clay Acrisol soils (Kotowska et al., 2015). b) despite higher tree stem density, basal area and root mass (all together maybe indicative of potential differences in nutrient demands of vegetation between these soils) in the loam than in the clay Acrisol soils (Appendix Table A2), the loam Acrisols still showed generally larger leaching fluxes than the clay Acrisols. c) based from the rubber plantations' (which all have the same low degree of management, e.g. unfertilized, in both soil types) Na leaching fluxes (the element more prone to leaching because of its monovalence and large hydration radius), the loam Acrisol soil was also higher ($P = 0.06$) than the clay Acrisols.

Previously we did not include these above points in the Discussion to keep the information more focused, especially that we have a data-rich paper. However, in the revised version we will include point (a) to address the reviewer's suggestion. We will

not point this additional discussion on the tree species composition but on the net primary production and vegetation structure, as these parameters are commonly used as indicators of vegetation's nutrient demand.

Author's changes in the manuscript: In the Discussion section 4.2, 2nd to the last sentence of the first paragraph, we added this:

Nutrient demand of vegetation may not be the dominant control on leaching fluxes, as the net primary production of these reference land uses did not differ between the loam and clay Acrisol soils (Kotowska et al., 2015). Similarly, the vegetation structure of the reference land uses (tree density, basal area, root biomass; Appendix Table A2) even seemed larger in the loam than the clay Acrisols.

3. L525-527 erosion and enhanced microbial mineralization of the native SOM can also contribute to low SOC stocks in oil palm plantation.

Author's response:

Yes, and we did not include in our discussion the erosion effect. Soil respiration in our oil palm plantations has significantly decreased compared to the reference land uses (Hassler et al., 2015). Based on correlations analysis with other soil biochemical parameters (15N signatures, SOC, P and base cation stocks), we attributed the reduced soil $CO_2$ fluxes from the oil palm plantations as the result of the strongly decomposed soil organic matter and reduced soil C stocks, which in turn are due to reduced litter input as well as to a possible reduction in C allocation to roots because of addition of nutrients from liming and P fertilization (Hassler et al., 2015).

Author's changes in the manuscript: Considering that erosion is an important process contributing to a decrease in SOC, we replaced the word 'strong' with 'additional' in this L525-527:

Moreover, the increased annual DOC fluxes in fertilized areas of oil palm plantations (Table 4) suggests a reduction in the retention of DOC in the soil. This, combined with

the decreases in litterfall and root production, harvest export (Kotowska et al., 2015), and decreases in soil CO2 emissions (Hassler et al., 2015) from the same oil palm plantations, provided additional support for the decreases in soil organic C stocks in smallholder oil palm plantations in the same study region (van Straaten et al., 2015).

4. L505-506 What data can support this statement?

Author's response: This statement (i.e., increases in dissolved Al and acidity of soil solution; Table 3) was based on the Results section where dissolved Al and soil water pH were presented (L348-349).

Author's changes in the manuscript: In this sentence, we now inserted Table 3 as the data source.

5. Table A2 sp. or spp. should not be written in italic. Dipterocarpaceae spp. include Shorea spp. The tree composition should be re-checked.

Author's response:

We agree with the reviewer. This Appendix Table A1 was based on information we had in 2015, when another group working on species diversity in our plots were yet continuing to identify the species, and we had by mistake doubly listed the families Dipterocarpaceae and Shorea. We will check the most common tree families (>20 individuals per plot) with the recent data of identified trees.

Author's changes in the manuscript:

We now not write sp. or spp. in normal font, deleted Shorea, will recheck these most common families of trees with the data of Rembold et al. (2017), and update this reference in the Table A1 instead of the previous unpublished data.

Rembold, K., Mangopo, H., Tjitrosoedirdjo, S.S., and Kreft, H.: Plant diversity, forest dependency, and alien plant invasions in tropical agricultural landscapes, Biodivers. Conserv., 213, 234-242, https://doi.org/10.1016/j.biocon.2017.07.020, 2017.

6. Throughout the paper, "l-1" and "L-1" are used inconsistently. Please use terms consistently.

Author's response: We appreciate very much for this very thorough read, and we indeed overlooked l-1.

Author's changes in the manuscript: We corrected the entire text to have uniformed unit abbreviation, L-1.

---

## Author Response (AR1)

Dear Prof. Dr. Frank Hagedorn,

On behalf of my co-authors, we extend our sincere gratitude for your helpful reviews and in facilitating the review of our manuscript, bg-2018-221. We have now incorporated all the changes we stipulated in our answers to the reviewers' comments. Additionally, from your suggestion:

Please also incorporate 'tree species' more explicitly into the final version of the manuscript. You nicely clarified that 'nutrient demand of vegetation and vegetation structure of the reference land-use are not the dominant control on leaching fluxes', but the reviewer asked for 'tree species'. I am aware that tree species and 'vegetation' are interlinked but you should clearly address/write that you do not expect that the different tree species composition between the sites was of minor importance for the leaching fluxes.

we have also addressed this in L443-449 of the revised manuscript, as also mentioned in our answers to Reviewer 2's specific comment #2.

For ease in reference, our answers to the reviewers' comments are now provided with the line numbers where the changes in our revised manuscript are reflected. All the line numbers are based on the revised manuscript with tracked-change and simple markup.

We hope that our revisions will satisfy your and the reviewers' questions and the standards of Biogeosciences. We look forward to hearing back from you. If there are any questions regarding our manuscript, I would be happy to clarify.

Sincerely yours,

Marife D. Corre

Point-by-point response to the reviews and a list of all relevant changes made in the manuscript

RC1 – Dr. Yit Arn Teh

Author's response:

First, we greatly appreciate the very detailed comments and suggestions of Dr. Teh. These help improve the clarity and broaden the perspective of our Discussion. All the comments of Dr. Teh are incorporated into our revisions. We describe below how we have addressed his comments. All the line numbers we referred to in our revision are based on the line numbers when tracked-change with simple markup is used (please note that line numbers change if the version of tracked-change with all markup is used).

General remarks and suggestions:

1. First, I think it may be worthwhile re-organizing the information in the discussion around the major findings, listing the top-level or most important findings first. The current structure of the discussion generally follows the order in which the results are reported, but there could be some value in arranging information according to the most ground-breaking or high impact results, in order to maximise the impact of the most important findings on the reader. Structuring a discussion in this way can be especially effective for data-rich papers like this one, because the discussion sections for data-rich papers can sometimes become quite large and extensive, and it is possible for key messages to get lost due to the volume of information covered.

Author's response:

We agree to this suggestion.

Author's changes in the manuscript:

We restructured the Discussion in the following:
Section 4.2– focuses first on the reference land uses.

L430-434, 452-453 - We put topic sentence in the beginning of the paragraphs to put first the high-impact results (previous L428-432 is condensed into a topic sentence; previous L447-448 is revised into a topic sentence). We minimized referencing back to the tables or figures with specific parameters, unless necessary. We streamlined every sentence to limit to the most convincing Results (previous L443-446 is deleted).
L449-451 - At the end of the paragraph, we give a take-home message (previous L458-460 is moved up to this place).

Section 4.3 – now titled land-use change effects.
The first paragraph focuses on the unfertilized rubber plantations and the second paragraph on the fertilized oil palm plantations, instead of separating the latter in the previous section 4.4. Although we emphasized in the previous manuscript version that the smallholder rubber plantations are not fertilized (the common practice of the smallholders in the area, at least during our study years of 2012-2013), we now highlighted this main difference in the soil management of the rubber and oil palm plantations (L469-471, L489-490).

L484-488, L512-514 - We also emphasized that the decreases in leaching losses in this rubber plantations and the increases in leaching losses in fertilized area of the oil palm plantations, as compared to the reference land uses, were mainly due to their management practices (i.e. without and with soil amendments). Again in each paragraph, we put a topic sentence (previous L465-470 is condensed into a topic sentence) to highlight the most important Results, and at the end of the paragraph a condensed take-home message (L525-528).

2. Second, another topic that is theoretically interesting and also policy-relevant is whether or not the investigators believe that over-fertilization is occurring for the rubber and oil palm systems? To phrase this another way, are the higher nutrient losses for rubber and oil palm because fertilizer inputs exceed plant/ecosystem demand, or because of the transport-reaction properties of the different soil types (e.g. do the exchange properties and rate of physical transport through the soil mean that the soil exchange complex cannot retain some of the added nutrients)? If the answer is the former, then this suggests that growers could be reducing their inputs of some elements. If the answer is the latter, then mitigation options become more complex, because they may require new means of introducing fertilizers to the soil (e.g. slow release fertilizers, organic fertilizers, soil conditioners to enhance CEC, etc.). It would be interesting if the investigators could expand upon this topic further in the discussion.

Author's response:

These are very good suggestions. We take into considerations all these points. Our discussion was geared both on the regulation of soil texture and fertilization rates. We focused on the absence of fertilization (rubber) and low (clay Acrisol, oil palm plantations) versus relatively high fertilization rates (loam Acrisol, oil palm plantations). We did not hone our discussion on exceedance of the nutrient retention capacity of the soils. The reason is because we had 2-5 times lower fertilization rates at our studied smallholder oil palm plantations than the nearby large-scale oil palm plantations, such that it will be very speculative to say that the soil exchange complex are saturated or increasingly unable to retain the added nutrients.

Author's changes in the manuscript:

L469-471 - We emphasized in the revised section 4.3 that the smallholder rubber plantations were not fertilized (at least during our study period of 2012-2013), and this was in part because the price of rubber had gone down at that time (as shown in the Supplementary Fig. 9 of Clough et al. 2016).

L494-497 - In the second paragraph of section 4.3, we also emphasized that the fertilizer application in oil palm plantations, despite at very low rates, resulted in increased nutrient leaching losses compared to the reference land uses, particularly in the loam Acrisol soil.

L529-535 - We stressed in the 4th paragraph of section 4.3 that the higher rates of fertilizer application in large-scale plantations than in smallholders imply for a need to optimize fertilization rate in order to minimize environmental effect while maintaining production level.

We also revised the previous L554-556 in Conclusion to this:
Management practices to regulate leaching losses are possibly more pressing for large-scale oil palm plantations, which have 2-5 times higher fertilization rates and may have a larger impact on ground water quality than the smallholders (L550-555). Process-based models, used to predict yield and associated environmental footprint of these tree cash crop plantations, should reflect the differences in soil management (e.g., absence or low vs. high fertilization rates, weed control) between smallholder and large-scale plantations (L555-558).

3. Third, two aqueous fluxes not included in this study are throughfall and stemflow. This observation is not meant as a criticism per se, as I fully recognize that this was very comprehensive and in-depth study, and resources are always limited for large-scale field experiments like this one. However, it would be useful if the investigators could comment on whether they think that differences in throughfall and stemflow among the different land-uses could have resulted in differences in nutrient dynamics and loss? Throughfall and stemflow are potentially influenced by factors such as vegetation structure (e.g. plant density), leaf area and tissue chemistry, so it is possible that the different cover types (with different vegetation structure and properties) could have different patterns in throughfall and stemflow, with knock-on effects for soil nutrient dynamics.

Author's response:

This is a very important point. We incorporated our views on this aspect in section 4.1 last paragraph. In terms of magnitude, the highest throughfall nutrient depositions (from peat soils, influenced by land-clearing fires in Kalimantan; Ponette-Gonzales et al., 2016) are still much lower (<1-3%) than the extant soil-N cycling rates and nutrient stocks in the top 0.1-m soil at our sites. The effects of atmospheric nutrient deposition may not be on how much this has added to the soil nutrient levels but on whether or not the receiving ecosystem can serve as a sink and is able to buffer its other cascading effects (e.g. acidification).

Author's changes in the manuscript:

To keep the manuscript in the same length, previous L411-417 was shortened into one sentence. To incorporate these points raised by Dr. Teh above, we replaced the previous L418-425 with this:

L419-427 - From a peatland site in Kalimantan, influenced by land-clearing fires, throughfall nutrient depositions (19-22 kg N, 6-11 kg P, 25-44 kg S ha$^{-1}$ yr$^{-1}$) are larger than those from bulk precipitation, indicating large contribution from dry deposition (Ponette-Gonzales et al., 2016). Total (dry + wet) nutrient depositions in our study region could be larger than the values from bulk precipitation. Such high atmospheric nutrient deposition may have fertilizing or polluting effect, depending on whether or not the receiving ecosystem is a sink and is able to buffer its other cascading effects (e.g., acidification). Additionally, atmospheric redistribution of nutrients in areas with widespread land-use conversion and intensification may have unforeseen effects on down-wind and down-stream ecosystems (e.g., Bragazza et al., 2016; Sundarambal et al., 2010).

These new references are added in the reference list:

Bragazza, L., Freeman, C., Jones, T., Rydin, H., Limpens, J., Fenner, N., Ellis, T., Gerdol, R., Hájek, M., Hájek, T., Iacumin, P., Kutnar, L., Tahvanainen, T., and Toberman, H.: Atmospheric nitrogen deposition promotes carbon loss from peat bogs, Proc. Natl. Acad. Sci. U.S.A., 103, 19386-19389, https://*doi*.org/10.1073/pnas.0606629104, 2006.

Ponette-González, A.G., LisaMCurran, L.M., Pittman, A.M., Carlson, K.M., Steele, B.G., Ratnasari, D., Mujiman, and Weathers, K.C.: Biomass burning drives atmospheric nutrient redistribution within forested peatlands in Borneo, Environ. Res. Lett., 11, 085003, https://doi.org/10.1088/1748-9326/11/8/085003, 2016.

Sundarambal, P., Balasubramanian, R., Tkalich, P., and He, J.: Impact of biomass burning on ocean water quality in Southeast Asia through atmospheric deposition: field observations, Atmos. Chem. Phys., 10, 11323–11336, https://doi.org/10.5194/acp-10-11323-2010, 2010.

SPECIFIC COMMENTS:

1. Lines 49-51: Provide information for wider context: It is worthwhile emphasizing here that smallholdings are very common through SE Asia, and account for approximately 40% of the land under production throughout the region. Therefore, while the smallholdings in Jambi may represent a larger proportion of land area than elsewhere in SE Asia, smallholdings are common and thus important to understand.

Author's response:

We incorporated this suggestion.

Author's changes in the manuscript:

L51-53 - The expansion of rubber and oil palm plantations has increased the income of Jambi, in particular the smallholder farmers (Clough et al., 2016; Rist et al., 2010), which account 99 % of rubber and 62 % of oil palm landholdings in the Jambi Province. In the whole of Indonesia, 85 % of rubber and 40 % of oil palm plantations are smallholders (DGEC, 2017).

DGEC (Directorate General of Estate Crops), Tree crop estate statistics of Indonesia 2015-2017: Palm oil and rubber, Indonesian Ministry of Agriculture, 2017, http://ditjenbun.pertanian.go.id/tinymcpuk/gambar/file/statistik/2017/Kelapa-Sawit-2015-2017.pdf, …/Karet-2015-2017.pdf.

2. Line 147-150: Consider re-phrasing the description of the fertilization rates, as the current wording makes it a bit more difficult to understand. One option may be to breakup this sentence into two shorter sentences; one referring to the clay Acrisol and the other to the loam Acrisol.

Author's response:

We take this in our revision.

Author's changes in the manuscript:

L150-152 - Fertilization rates were 48 kg N, 21 kg P and 40 kg K $ha^{-1}$ $yr^{-1}$ in the clay Acrisol soil, whereas these were 88 kg N, 38 kg P $ha^{-1}$ $yr^{-1}$ and 157 kg K $ha^{-1}$ $yr^{-1}$ (accompanied by Cl input of 143 kg Cl $ha^{-1}$ $yr^{-1}$) in the loam Acrisol soil.

3. Line 161: Minor question or point for clarification: Do the authors have any insight as to where nutrient-acquiring roots proliferate in this system? Is it possible that sampling 1.3-1.5 m from the palm could slightly overestimate the rate of leaching loss? Oil palms tend to show the highest density of roots within 1 m of the plant stem; therefore, it is possible that by sampling outside of this region the investigators may underestimate plant uptake or overestimate leaching. Arguably, however, it is not clear if all the roots within 1 m of the palm stem are active or specialized for nutrient uptake, i.e. many of these roots may be dead or not directly involved in nutrient acquisition. Moreover, if the growers' practice is to apply fertilizer 1.3-1.5 m from the stem, then it is likely that this sampling scheme is likely to best represent actual trends in leaching. It is also possible that the roots produced 1.3-1.5 m from the stem are tracking nutrient availability and are specialized for nutrient uptake.

Author's response:

From another study (conducted by another group in this collaborative research center) that measured root distribution in the same smallholder oil palm plantations, there were no significant correlations between root mass distribution with distance to palms. This was attributed to the facts that these are mature plantations (12-16 yrs old, except one site that was 9 yrs old) and the weeding practices in smallholder plantations were not intensive (2 times per year only) and hence the ground was almost always covered with undergrowth. We think that we did not over-represent the leaching losses from the fertilized area as these values were averaged with the leaching losses from under the frond stacks to get a plot-scale estimate for an oil palm plantation.

4. Line 163: Did the growers plant any understory plants for erosion control? If so, did the authors sample from these areas too? Although the biomass and uptake capacity of these herbaceous plants is likely to be low relative to mature palms, leaching patterns are likely to be different from unvegetated areas.

Author's response:

In these smallholder oil palm plantations, the ground vegetation was that from natural regrowth after the 2-times weeding per year. The ground was mostly covered with understory plants for most part of the year.

5. Lines 268-279: Do the investigators have an estimate for the nutrient input from throughfall and stemflow? If these data do not exist, is it possible to constrain these values in the model from similar systems? While rain water provides a useful end-member with which to estimate the nutrient content of "external" moisture inputs, it is possible that dry deposition of nutrients and leaching from aboveground plant parts could contribute to the nutrient input to soil. Especially if this region is near local sources of N pollution, it is possible that throughflow/stemflow could make a contribution to the overall N load to the soil.

Author's response:

We did not measure stem flow and throughfall, and there are no data for stem flow and throughfall for these land uses for Jambi or Sumatra that we are aware of. It is likely that stem flow + throughfall is larger than from bulk precipitation in areas with large dry deposition from biomass burning. In terms of magnitude, those high throughfall nutrient depositions from heavily fire-impacted peatlands in Kalimantan (Ponette-González et al., 2016) are still much lower than extant N cycling rates and macronutrient stocks in the soil at our sites. As nutrients deposited into a system will eventually be incorporated into the soil-plant cycling, changes in leaching losses are ultimately reflecting how efficient the system (soil, biota and vegetation) is in retaining the nutrients from both external sources and internal cycling.

Author's changes in the manuscript:

Please see our answers to reviewer's major comment 3 above.

6. Lines 317-320: What are the comparable values for ET, run-off and drainage for rubber and oil palm systems?

Author's response:

We provide this information.

Author's changes in the manuscript:

L320-322 - In rubber and oil palm, modelled annual ET was 30-32 %, runoff was 22-31 %, and drainage was 37-47 % of annual precipitation.

7. Lines 395-534: Given that the authors introduce testable hypotheses in the introduction, I think it's important to "close the circle" by referencing these hypotheses in the discussion, and confirming if the authors' findings supported or falsified their hypotheses.

Author's response:

We agree with the reviewer.

Author's changes in the manuscript:

We re-structured the Discussion according to the 1st and 2nd major suggestions above. In the Conclusion, we closed the circle by linking our findings to our hypotheses (L540-544).

8. Line 441: Further clarification required re: the phrase "higher rates of soil NH4+ cycling." For those who have not yet read Allen et al. (2015), does this phrase mean that the rate of NH4+ mineralization is greater, gross production and uptake of NH4+ is greater, or that the overall turnover of NH4+ is greater?

Author's response:

Gross production and immobilization of NH4+ were positively correlated and both rates are large, meaning this internal cycling was large and closely-coupled, and was mirrored by low TDN leaching fluxes in the clay than loam Acrisol soils.

Author's changes in the manuscript:

We changed this sentence accordingly (L441-442).

9a. Lines 491-534: One question and one comment: first, given the finding that fertilization is enhancing leaching losses in these smallholder landscapes, do the authors believe that the growers are over-fertilizing? Is the high rate of leaching loss because the plant demand is lower than nutrient supply, or is it because transport factors mean that the nutrients are lost before plants are able to take-up the nutrients? The authors expert assessment directly influences policy and management decisions; if it is an over-fertilization situation (i.e. plant demand « nutrient input), then the mitigation option would be to reduce fertilizer inputs. If it is an issue of transport (e.g. ion exchange sites are saturated or movement of soil solution is too rapid for efficient plant uptake), the different mitigation options suggest themselves (e.g. use of slower release fertilizers, or other technologies to reduce nutrient transport through the soil column). The conclusion that soil texture was the dominant influence (lines 528-529) tends to imply that the authors believe the second option is more likely (i.e. rapid transport leads to loss, rather than plant demand « nutrient input); however, it would be useful to hear the authors thoughts on this topic given its wider importance for mitigation of nutrient pollution.

Author's response:

These smallholders are not over-fertilizing; please see also our answer to the 2nd major comment above. The high leaching losses from oil palm plantations occurred particularly on the fertilized area around the palm base and were higher in the loam Acrisol soil, which also happened to have a larger fertilization (e.g., 48 vs. 88 kg N/ha/yr, section 2.1), than the clay Acrisol soil. The plant demand, using simply the index of fruit harvest export (72-96 kg N/ha/yr; Kotowska et al. 2015), was certainly higher than the fertilization rates of the smallholders, whose low fertilization rates were largely determined by their resources of being able to afford the cost of fertilizers. Thus, our discussion was focused more on the regulation of soil texture and fertilization rate rather than on the exceedance of the ecosystem's capacity to retain the added nutrients from fertilizers. The leaching losses that we measured are possibly contributed by both soil texture (not only through adsorption/exchange capacity but also on solute transport) and fertilizer application. We think the role of transport occurred on pulses or time periods when high rainfall occurred following the 2 periods in a year when fertilizers were applied. Unlike, however, $N_2O$ emissions from the soil surface of which temporal pattern following fertilization are clearly manifested (Hassler et al. 2017), this is not the case for leaching losses - probably because transport of solute down to 1.5-m depth (lysimeter sampling) can take days of intermittent rainfall events.

Author's changes in the manuscript:

As to the aspect of our findings' implications to management, we addressed this in the last paragraph of section 4.3.

L529-535 - The fertilization rates in our studied smallholder oil palm plantations were only 2-5 times lower than the nearby large-scale plantations, typically with 230-260 kg N ha$^{-1}$ yr$^{-1}$. Our findings, that leaching of TDN and base cations increased and their retention efficiency decreased particularly in the loam Acrisol despite the low fertilization rates (Tables 4 and 5), imply for a need to optimize fertilization rate in large-scale plantations, especially on coarse-texture soils which have low inherent nutrient retention, in order to minimize environmental effect while maintaining production.

9b. My second point is a comment rather than a question. One of the challenges in predicting the behaviour of smallholder systems is that there is potentially a wider diversity of practices and fertilization schemes compared to large-scale industrial plantations. For instance, depending on the relative wealth or resources of individual growers, they may have better access to fertilizers than less fortunate growers. While this does not necessarily take away from the message that the authors are trying to convey here (i.e. that certain types of more "intensive" or "invasive" land-use can show enhanced leaching losses), I think it is useful to discuss this source of potential variance and uncertainty, since it means that we have to develop better process-based models so that we can adequately predict flux from smallholder systems.

Author's response:

We completely agree to this comment.

Author's changes in the manuscript:

We added this in the Conclusion.

L555-558 - Process-based models, used to predict yield and associated environmental footprint of these tree cash crop plantations, should reflect the differences in soil management (e.g., absence or low vs. high fertilization rates, weed control) between smallholder and large-scale plantations.'

10. Table and figure legends: Minor pedantic point: throughout the table and figure legends, the authors refer to the loam Acrisol and clay Acrisol as two different "landscapes." While I do not consider this as problematic as such, I wonder if the phrase "soil orders" or "soil types" may be more intuitive for the reader, given that the reference for these two types of environments are the names of the soil orders?

Author's response:

Yes, we agree. In the revised version, we changed all these in the text from landscapes to soil types.

RC2 – Dr. K. Fujii

Author's response:

First, we extend our sincere thanks to Dr. Fujii for his insightful suggestions and comments that help improve our manuscript greatly. We describe how we have addressed his comments in our answers below. All the line numbers we referred to in our revision are based on the line numbers when tracked-change with simple markup is used (please note that line numbers change if the version of tracked-change with all markup is used).

1. One of major issues is soil classification (Acrisols). Sumatra soils are more or less affected by volcanic ash deposition. Soils are relatively young among Indonesian soils. I am afraid whether the soils studied satisfy Acrisols' low clay activity. Please confirm the soil profile data especially in the Bt horizon. Low CEC/clay is required. In addition, both loam and clay Acrisols contain high contents of clays. Please clarify how two types were separated.

Author's response:

Soil particle size distribution was determined from the three sites (or replicate plots) of each land use within each landscape (which was subsequently classified according to the major soil texture group). In each site, soil samples for particle size analysis was taken from 6 depth intervals within 2-m depth. For the general soil texture classification, we took the values of depth-weighted average for each land-use site, and then the averaged for the 12 land-use sites (4 land uses x 3 sites) for each landscape. Similarly, cation exchange capacity (CEC) was determined from 2 to 5 samples per depth for each land-use site. We also did an oxalate-extraction for Fe and Al, and these characteristics did not satisfy for an Andic property.

Our soil texture classification is based on the averages of the plots in order to come up with a general category of soil texture. The clay area had an average across sites of 48% clay, 27% silt and 25% sand. The loam area had particle size fractions that bordered between loam and (sandy) clay loam, and for ease in writing the classification in the manuscript, we generally termed this as loam (22-32% clay, 25-30% silt and 45-50% sand).
In the clay soils (with more than 40% clay), >8% increase in clay was observed in the subsoil. In the loam soils, the ratio of subsoil % clay to overlying layer was >1.2. Both of these metrics satisfied the criterion for an Argic horizon. These Argic horizons of our sites has CEC of < 24 $cmol_c$ $kg^{-1}$ clay and base saturation of <10% (all reported in our earlier publication, Allen et al. 2016).

Author's changes in the manuscript:

As these characteristics were already reported in our earlier publication (Allen et al. 2016), we did not elaborate these in the present manuscript.
However, in order to address this concern of Dr. Fujii, we inserted in section 2.1, after a short description of the loam and clay Acrisol soils, the following:

L122 - Detailed soil characteristics of these classifications are reported by Allen et al. (2016).

2. L447 The authors link between N leaching and the acid-buffering capacity of the soils, but link between N leaching and clay contents will be precise. Exchangeable Al as well as pH is a record of soil acidification, but quantitative link between N loss and soil ANC cannot be supported by calculating proton budgets in soil.

Author's response:

We agree with the reviewer's comments. As stated in the Results (previous L366-376 of the original manuscript), N leaching fluxes (and N and base cation retention efficiency) were significantly correlated with soil base saturation, ECEC and organic C which were, in turn, correlated with clay content. Thus, this L447 is a leap in our interpretation, and we changed this L447 (as a topic sentence for this Discussion on correlations of nutrient leaching fluxes with soil biochemical characteristics) to refer only to what were clearly reflected by the correlation tests.

Author's changes in the manuscript:

We changed L447 to:

L452-453 - The influenced of soil texture on soil biochemical characteristics also linked to the leaching losses or, conversely, nutrient retention efficiency.

3. There were some droughts or dry-wet cycles in Indonesia. This has strong impacts on solute concentration and leaching flux. I recommend to add correlation analyses between water flux (or soil water content) and solute concentration to check dilution or condensation effects by dry-wet cycles. This effect can affect annual nutrient loss as well. At least, adding discussion will improve manuscript.

Author's response:

From the start of our data analyses, we have explored all correlation tests, including correlations of element concentrations with the modelled soil moisture content at 1.5-m depth (lysimeter sampling) in order to check for dilution or condensation; the correlation tests with soil moisture contents were conducted in a similar manner as those sreported in Appendix Tables 3 and 4. The only significant correlation coefficients with soil moisture contents were found for the fertilized area of the oil palm plantations in the loam Acrisol soil for K, Ca, Mg and total S concentrations (r = -0.59 – -0.72, P ≤ 0.05, n = 12 monthly measurements for one year). However, the influence of soil water content on nutrient concentrations were anyway incorporated in the calculation of leaching fluxes (nutrient concentration x drainage flux), as stated in L247-250. Also, all the correlations that used the nutrient concentrations (Appendix Tables 3 and 4) were interpreted only to assess which cations were correlated with which anions (i.e., L340-342, L350-352, L363-366) in order to support the partial ionic charge balance of solutes, as depicted in Fig. 1.

Author's changes in the manuscript:

Considering that (1) these above correlation tests were only significant in four elements at one spatial category (fertilized area of oil palm in the loam soils) and (2) the influence of soil water content on element concentrations were incorporated in the calculation of leaching fluxes, we will not add this in the Discussion. This is so that the main highlights of our findings will not be buried, as suggested by Dr. Teh's (reviewer 1) first major suggestion.

4. Throughout the paper, the authors use the ambiguous term "soil fertility". The definition of soil fertility is not same among the readers. Please define it in the beginning of the paper. Most of soil scientists avoid to use the term "soil fertility" in scientific paper.

Author's response:

We agree and take this suggestion. When we used the word soil fertility, we specified in parenthesis the soil biochemical characteristics that we used as basis. These were already done in the original manuscript version of the Introduction and M&M.

Introduction L61-62 - Soil texture affects nutrient leaching through its control on soil fertility (e.g., cation exchange capacity, decomposition, and nutrient cycling) and soil water-holding capacity.

M & M L123-125 - In summary, the soil textural difference leads to inherent differences in soil fertility (e.g., higher effective cation exchange capacity, base saturation, Bray-extractable P and lower Al saturation) in the clay than the loam Acrisols under forest and jungle rubber (Appendix Table A1).

Author's changes in the manuscript:

We specified the soil biochemical properties when we used the word *soil fertility*.
Namely:
Discussion, section 4.2, the last sentence of 1$^{st}$ paragraph:
L449-451 - Our findings showed that soil texture was the main factor regulating nutrient leaching losses and soil fertility (e.g., nutrient stocks and N-cycling rates) in these highly weathered Acrisol soils.

In the Abstract and Discussion, section 4.3 last sentence of 1$^{st}$ paragraph – we replaced '*soil fertility*' with '*nutrient availability*':
L484-488 - Our results showed that disruption of nutrient cycling between the soil and vegetation brought about by land-use conversion to rubber plantations, combined with the absence of soil amendments, had decreased nutrient leaching (Tables 3 and 4) as well soil nutrient availability (i.e., P stocks, microbial N, gross N mineralization rates; Allen et al., 2015; Allen et al., 2016).

In the Conclusion, we replaced *soil fertility* with *soil nutrients* or *nutrient levels*.

Specific comments:

1. The authors regarded jungle rubber as original vegetation, but it is introduced from Brazil some hundreds of years ago. It is not native vegetation.

Author's response:

We replaced the word '*original*' with '*previous*' (in the Introduction) or '*reference land use*' (in the Discussion), which was what we actually meant. We use '*previous*' or '*reference land use*' to denote the land use immediately before the conversion to rubber and oil palm.

2. The authors ascribed the greater nutrient losses from loam Acrisols than those from clay Acrisols. However, tree composition is not same between two sites. The authors need to add careful discussion on this topic.

Author's response:

This is a very good point and we take this into consideration. Our interpretation that soil texture was the main factor influencing leaching losses is based on the following:
a) total net primary production (aboveground + belowground), as an indicator of plant usage of soil nutrients, of the forest and jungle rubber did not differ between the loam and clay Acrisol soils (Kotowska et al., 2015).
b) despite higher tree stem density, basal area and root mass (all together maybe indicative of potential differences in nutrient demands of vegetation between these soils) in the loam than in the clay Acrisol soils (Appendix Table A2), the loam Acrisols still showed generally larger leaching fluxes than the clay Acrisols.
c) based from the rubber plantations' (which all have the same low degree of management, e.g. unfertilized, in both soil types) Na leaching fluxes (the element more prone to leaching because of its monovalence and large hydration radius), the loam Acrisol soil was also higher (P = 0.06) than the clay Acrisols.

Previously we did not include these above points in the Discussion to keep the information more focused, especially that we have a data-rich paper. However, in the revised version we will include points (a) and (b) to address the reviewer's suggestion.

Author's changes in the manuscript:

In the Discussion section 4.2, we added this:

L443-449 - Nutrient demand of vegetation may not be the dominant control on leaching fluxes, as the vegetation structure of the reference land uses (tree density, basal area, root biomass; Appendix Table A2) even seemed larger in the loam than the clay Acrisols. Similarly, the differences in tree species compositions between the loam and clay Acrisol soils (Appendix Table A2) may not have influenced the nutrient leaching fluxes, as supported by the comparable net primary production of the reference land uses between soil types (Kotowska et al., 2015).

3. L525-527 erosion and enhanced microbial mineralization of the native SOM can also contribute to low SOC stocks in oil palm plantation.

Author's response:

Yes, and we did not include in our discussion the erosion effect. Soil respiration in our oil palm plantations has significantly decreased compared to the reference land uses (Hassler et al., 2015). Based on correlations analysis with other soil biochemical parameters ($^{15}$N signatures, SOC, P and base cation stocks), we attributed the reduced soil $CO_2$ fluxes from the oil palm plantations as the result of the strongly decomposed soil organic matter and reduced soil C stocks, which in turn are due to reduced litter input as well as to a possible reduction in C allocation to roots because of addition of nutrients from liming and P fertilization (Hassler et al., 2015).

Author's changes in the manuscript:

Considering that erosion is an important process contributing to a decrease in SOC, we replaced the word '*strong*' with '*additional*':

L519-525 - Moreover, the increased annual DOC fluxes in fertilized areas of oil palm plantations (Table 4) suggests a reduction in the retention of DOC in the soil. This, combined with the decreases in litterfall and root production, harvest export (Kotowska et al., 2015), and decreases in soil $CO_2$ emissions (Hassler et al., 2015) from the same oil palm plantations, provided *additional* support for the decreases in soil organic C stocks in smallholder oil palm plantations in the same study region (van Straaten et al., 2015).

4. L505-506 What data can support this statement?

Author's response:

This statement (i.e., increases in dissolved Al and acidity of soil solution; Table 3) was based on the Results section where dissolved Al and soil water pH were presented (L352-354, L360-362).

Author's changes in the manuscript:

In this sentence (L502-504), we now inserted Table 3 as the data source.

5. Table A2 sp. or spp. should not be written in italic. Dipterocarpaceae spp. include Shorea spp. The tree composition should be re-checked.

Author's response:

We agree with the reviewer. This Appendix Table A1 was based on information we had in 2015, when another group working on species diversity in our plots were yet continuing to identify the species, and we had by mistake doubly listed the families Dipterocarpaceae and Shorea. We now checked the 5 numerous tree families with the most current identified trees.

Author's changes in the manuscript:

We updated the 5 numerous tree families in Appendix Table A2 based from Rembold et al. (2017) and Rembold (pers. comm.).

Rembold, K., Mangopo, H., Tjitrosoedirdjo, S.S., and Kreft, H.: Plant diversity, forest dependency, and alien plant invasions in tropical agricultural landscapes, Biodivers. Conserv., 213, 234-242, https://doi.org/10.1016/j.biocon.2017.07.020, 2017.

6. Throughout the paper, "l-1" and "L-1" are used inconsistently. Please use terms consistently.

Author's response:

We appreciate very much for this very thorough read, and we indeed overlooked $l^{-1}$.

Author's changes in the manuscript:

We corrected the entire text and Table 3 to have uniformed unit abbreviation, $L^{-1}$.

Marked-up manuscript version

[revised manuscript text omitted]

(mg Mg $L^{-1}$)

| | | | | | |
|---|---|---|---|---|---|
| Total aluminum | 0.4 (0.1) b A | 0.2 (0.0) c | 0.3 (0.0) b | 1.2 (0.7) a | 0.1 (0.0) c |
| (mg Al $L^{-1}$) | | | | | |
| Total iron (mg Fe $L^{-1}$) | 0.2 (0.1) A† | 0.0 (0.0) | 0.0 (0.0) | 0.0 (0.0) | 0.1 (0.1) |
| Total manganese | 0.02 (0.00) | 0.01 (0.00) | 0.01 (0.00) | 0.01 (0.00) | 0.01 (0.00) B |
| (mg Mn $L^{-1}$) | | | | | |
| Total phosphorus | 0.008 (0.0) a† | 0.004 (0.0) b† | 0.003 (0.0) c† | 0.005 (0.0) ab† | 0.005 (0.0) ab† |
| (mg P $L^{-1}$) | | | | | |
| Total sulfur (mg S $L^{-1}$) | 0.16 (0.00) a† | 0.14 (0.00) bc† | 0.10 (0.00) c† | 0.14 (0.00) ab† | 0.12 (0.00) b† |
| Total silica (mg Si $L^{-1}$) | 0.5 (0.1) | 0.3 (0.1) B† | 0.2 (0.1) | 0.3 (0.1) | 0.2 (0.0) |
| Chloride (mg Cl $L^{-1}$) | 8.9 (0.8) b A† | 6.6 (0.8) c | 6.7 (0.6) c | 21.0 (2.7) a | 6.2 (0.8) c |

| clay Acrisol soil | | | | | |
|---|---|---|---|---|---|
| pH | 4.3 (0.1) c | 4.4 (0.1) bc | 4.4 (0.0) c | 4.6 (0.1) ab | 4.6 (0.1) a |
| Ammonium | 0.2 (0.0) B† | 0.1 (0.0) | 0.1 (0.0) | 0.2 (0.0) | 0.1 (0.0) |
| (mg $NH_4^+$-N $L^{-1}$) | | | | | |
| Nitrate | 0.1 (0.0) | 0.0 (0.0) B | 0.2 (0.1) | 0.9 (0.9) | 0.0 (0.0) |
| (mg $NO_3^-$-N $L^{-1}$) | | | | | |
| Dissolved organic N | 0.1 (0.0) a†B† | 0.1 (0.0) a† | 0.1 (0.0) ab† | 0.0 (0.0) b† | 0.0 (0.0) b† |
| (mg N $L^{-1}$) | | | | | |
| Total dissolved N | 0.3 (0.0) B† | 0.2 (0.0) B† | 0.4 (0.1) | 1.1 (0.9) | 0.2 (0.0) |
| (mg N $L^{-1}$) | | | | | |
| Dissolved organic C | 3.3 (0.4) | 4.0 (0.3) | 2.9 (0.1) | 4.8 (0.9) | 4.4 (1.1) |
| (mg C $L^{-1}$) | | | | | |
| Sodium (mg Na $L^{-1}$) | 2.4 (0.2) bc B | 2.5 (0.1) b | 2.0 (0.1) c | 4.6 (1.2) a | 2.5 (0.5) bc |
| Potassium (mg K $L^{-1}$) | 0.3 (0.0) | 0.3 (0.1) | 0.3 (0.0) | 0.4 (0.1) | 0.2 (0.1) |

| | | | | | |
|---|---|---|---|---|---|
| Calcium (mg Ca $L^{-1}$) | 0.7 (0.1) | 0.7 (0.0) | 0.7 (0.1) | 0.8 (0.2) | 0.5 (0.1) |
| Magnesium (mg Mg $L^{-1}$) | 0.3 (0.0) B | 0.3 (0.0) | 0.3 (0.0) | 0.4 (0.1) | 0.2 (0.1) |
| Total aluminum (mg Al $L^{-1}$) | 0.2 (0.0) B | 0.2 (0.1) | 0.3 (0.1) | 0.2 (0.1) | 0.1 (0.0) |
| Total iron (mg Fe $L^{-1}$) | 0.0 (0.0) b† B† | 0.0 (0.0) b† | 0.0 (0.0) b† | 0.0 (0.0) b† | 0.1 (0.0) a† |
| Total manganese (mg Mn $L^{-1}$) | 0.01 (0.00) | 0.01 (0.00) | 0.01 (0.00) | 0.08 (0.10) | 0.02 (0.00) |
| Total phosphorus (mg P $L^{-1}$) | 0.010 (0.0) | 0.004 (0.0) | 0.004 (0.0) | 0.004 (0.0) | 0.010 (0.0) |
| Total sulfur (mg S $L^{-1}$) | 0.15 (0.00) | 0.11 (0.00) | 0.11 (0.00) | 0.13 (0.00) | 0.12 (0.00) |
| Total silica (mg Si $L^{-1}$) | 0.4 (0.0) | 0.6 (0.1) A† | 0.3 (0.0) | 1.0 (0.4) | 0.7 (0.2) |
| Chloride (mg Cl $L^{-1}$) | 6.4 (0.6) B† | 6.8 (0.9) | 5.7 (0.8) | 7.2 (2.1) | 4.6 (0.8) |

**Table 4.** Mean (± SE, *n* = 4, except for oil palm *n* = 3) annual (2013) nutrient leaching fluxes measured at a depth of 1.5 m in different land uses within the loam and clay

Acrisol soils in Jambi, Sumatra, Indonesia. Means followed by different lowercase letters indicate significant differences among land uses within each  soil type and different uppercase letters indicate significant differences between  soil types for each reference land use (Linear mixed effects models with Fisher's LSD test at $P \le 0.05$, and † at $P$

$\le 0.09$ for marginal significance).

[revised manuscript text omitted]